# Intravenous administration of BCG in mice promotes natural killer and T cell-mediated antitumor immunity in the lung

Eduardo Moreo ®[1,2], Aitor Jarit-Cabanillas[3], Iñaki Robles-Vera[3], Santiago Uranga ®[1,2], Claudia Guerrero[1,2], Ana Belén Gómez[1,2], Pablo Mata-Martínez[4], Luna Minute[4], Miguel Araujo-Voces[1,2,5], María José Felgueres ®[6], Gloria Esteso ®[6], Iratxe Uranga-Murillo ®[7,8], Maykel Arias[7,8], Julián Pardo ®[7,8], Carlos Martín ®[1,2], Mar Valés-Gómez ®[6], Carlos del Fresno ®[4], David Sancho ®[3] & Nacho Aguiló ®[1,2] ✉

Intravesical administration of Bacillus Calmette-Guérin (BCG) was one of the first FDA-approved immunotherapies and remains a standard treatment for bladder cancer. Previous studies have demonstrated that intravenous (IV) administration of BCG is well-tolerated and effective in preventing tuberculosis infection in animals. Here, we examine IV BCG in several preclinical lung tumor models. Our findings demonstrate that BCG inoculation reduced tumor growth and prolonged mouse survival in models of lung melanoma metastasis and orthotopic lung adenocarcinoma. Moreover, IV BCG treatment was well-tolerated with no apparent signs of acute toxicity. Mechanistically, IV BCG induced tumor-specific CD8[+] T cell responses, which were dependent on type 1 conventional dendritic cells, as well as NK cell-mediated immunity. Lastly, we also show that IV BCG has an additive effect on anti-PD-L1 checkpoint inhibitor treatment in mouse lung tumors that are otherwise resistant to anti-PD-L1 as monotherapy. Overall, our study demonstrates the potential of systemic IV BCG administration in the treatment of lung tumors, highlighting its ability to enhance immune responses and augment immune checkpoint blockade efficacy.

Globally, lung cancer is the second most prevalent malignancy and is associated with the highest mortality rates, accounting for approximately 18% of all cancer-related deaths[1]. Moreover, the lung is a common site for the development of metastases originating from various primary tumors, including breast, colon, pancreas, skin, or kidney,

among others. The lung, being highly vascularized, is susceptible to the spread of cancer cells through the bloodstream or lymphatic system. Remarkably, the presence of lung metastases represents a significant bad prognostic factor associated with a drastic reduction in patient survival.

[1]Grupo de Genética de Micobacterias, Departamento de Microbiología, Pediatría, Radiología y Salud Pública, Facultad de Medicina, Universidad de Zaragoza, IIS-Aragon, Zaragoza, Spain. [2]CIBER Enfermedades Respiratorias, Instituto de Salud Carlos III, Madrid, Spain. [3]Centro Nacional de Investigaciones Cardiovasculares Carlos III (CNIC), Madrid, Spain. [4]Hospital la Paz Institute for Health Research (IdiPAZ), Madrid, Spain. [5]Departamento de Bioquímica y Biología Molecular, Instituto Universitario de Oncología (IUOPA), Universidad deOviedo, Oviedo, Spain. [6]Departamento de Inmunología y Oncología, Centro Nacional de Biotecnología (CNB-CSIC), Madrid, Spain. [7]Grupo de Inmunoterapia, Inmunidad y Cáncer, Departamento de Microbiología, Pediatría, Radiología y Salud Pública, Facultad de Medicina, Universidad de Zaragoza, IIS-Aragon, Zaragoza, Spain. [8]CIBER Enfermedades Infecciosas, Instituto de Salud Carlos III, Madrid, Spain. ✉e-mail: naguilo@unizar.es

In recent years, the development of immunotherapy-based treatments has marked a significant breakthrough in the fight against cancer. Immune checkpoint inhibitors (ICIs), specifically those targeting the PD-1/PD-L1 axis, have emerged as first-line treatments for advanced non-small cell lung cancer (NSCLC)[2,3] as well as lung metastases of different origins, such as skin, colorectal, breast or renal cell carcinoma[4–6]. However, a considerable proportion of patients with lung tumors (~70%) fail to benefit from ICIs[7,8], leaving them with limited therapeutic options. Mechanistically, the engagement of the PD-1 receptor on T cells hampers TCR signaling, resulting in the suppression of T cell proliferation and effector functions[9]. ICIs targeting the PD-1/PD-L1 axis disrupt this inhibitory interaction, reinvigorating T cell antitumor responses[10]. However, the failure of ICIs can be attributed to various factors within the tumor microenvironment (TME)[11,12]. The TME is often characterized by an immunosuppressive milieu, where several factors like immune inhibitory cells, cytokines, and metabolic changes impede effective antitumor immune responses driven by T cells[13,14]. Furthermore, dysfunction in critical immune cell populations, such as conventional dendritic cells (cDC1s) and natural killer (NK) cells, can contribute to tumor immune evasion and ineffectiveness of ICIs[15,16].

More than a century ago, Raymond Pearl observed a correlation between the presence of tuberculosis lesions in the lungs and lower ratios of cancer during autopsies[17,18]. Building upon these findings, researchers hypothesized over subsequent decades that the attenuated tuberculosis vaccine, bacille Calmette-Guérin (BCG), could be used as an anticancer therapy[19]. As a result, clinical trials have been conducted to investigate the effect of local administration of BCG against various tumor types such as melanoma, prostate or lung cancer. In the 1970s, the successful development of BCG application for bladder tumors led to its approval as one of the first immunotherapies by the FDA in 1990[20]. Indeed, more than four decades later, intravesical BCG is now a first-line therapy for high-risk non-muscle invasive bladder cancer (NMIBC) patients.

In recent years, intravenous (IV) BCG administration has emerged as a promising tuberculosis vaccine strategy in mice and non-human primates (NHPs)[21,22]. Studies in mice have shown that IV BCG spreads to different organs including the bone marrow (BM), spleen, liver or lungs, where it can persist for several weeks[22,23]. IV BCG induces profound alterations in immune populations from colonized organs. Notably, BCG impacts myelopoiesis in the bone marrow (BM) and elicits trained immunity in myeloid cell progenitors, resulting in the generation of mature macrophages with a "trained" phenotype. These macrophages exhibit quicker and more efficient responses upon subsequent pulmonary challenge with *Mycobacterium tuberculosis*[22].

In this study, we describe a therapeutic antitumoral approach based on a single IV inoculation of BCG, which proved to be well-tolerated and effective in various mouse models of metastatic and orthotopic lung tumors. Our findings unveil the mechanism behind the antitumor effect of IV BCG, highlighting the coordinated stimulation of both innate and adaptive immune responses. This study provides valuable insights into the interactions among different immune cellular subsets within the context of lung tumors, ultimately leading to an effective antitumor immune response.

## Results
### Therapeutic efficacy of intravenous BCG against metastatic lung tumors
Previous studies investigating IV BCG immunization as a tuberculosis vaccine have shown that after systemic inoculation, the bacteria can colonize various organs, including the lungs. This event triggers robust lung immune responses, conferring protection against pulmonary viral and bacterial infections, including SARS-CoV-2 or *Mycobacterium tuberculosis*[21,24]. Considering the successful use of BCG for bladder cancer, we sought to investigate whether immune system stimulation by IV BCG could effectively inhibit lung tumor growth in mice. To test

this idea, we initially used a well-known model of lung melanoma metastasis based on IV inoculations of the B16-F10 melanoma cell line, which generates lung metastasic nodules. Histological analysis of lung tissue sections revealed that metastatic nodules were already established 7 days after tumor cell inoculation and grew progressively at later time points (Supplementary Fig. 1). Tumor-bearing mice were treated with a single IV injection of BCG Pasteur ($10^6$ CFUs) at 7 days after tumor cell administration (Fig. 1a). IV BCG significantly extended mouse survival from a median of 26 days in PBS group to 38 after BCG treatment (Fig. 1b). Since the efficacy of immunotherapy can be compromised in higher tumor burden scenarios, we delayed treatment until day 14, a timepoint in which metastatic nodules were grossly visible (Supplementary Fig. 1). Our data revealed that the therapeutic benefit conferred by IV BCG was maintained under this more stringent setting (Fig. 1c).

As part of the efficacy analysis, we assessed the lung tumor burden at day 20 after tumor cell inoculation (and day 13 following BCG inoculation) through histological analysis. Our findings demonstrated that IV BCG reduced tumor burden, as measured by the proportion of lung area covered by tumors in lung tissue sections (Fig. 1d and Supplementary Fig. 2a). These findings were further confirmed by a clonogenic assay, in which lung single cell suspensions from BCG-treated tumor-bearing mice exhibited a lower capacity to form colonies ex vivo, consistent with the reduced tumor burden observed (Fig. 1e and Supplementary Fig. 2b).

Then, we compared IV administration with subcutaneous (SC) and intranasal (IN) delivery routes, previously characterized against tuberculosis disease. IN BCG was effective in reducing lung tumor growth, although to a lesser extent than the IV route (Fig. 1f). Conversely, SC BCG administration was completely ineffective (Fig. 1f), suggesting that direct contact of the vaccine with the lung microenvironment, achieved through the IN and IV routes[24,25], is necessary for BCG therapeutic effect. Additionally, we examined the dose-response profile of IV BCG and found that the $10^6$ CFU dose was more effective than $10^5$ (Fig. 1g), but there were no significant differences between the $10^6$ and $10^7$ doses. As a result, we selected the $10^6$ CFU dose for subsequent experiments. Finally, we also discovered that viable BCG was required for antitumoral efficacy, as administration of $10^7$ heat-killed (HK) BCG bacilli did not provide any survival advantage compared to the PBS-treated control group (Fig. 1g).

Considering potential safety concerns associated with the intravenous inoculation of a live-attenuated bacteria, we evaluated the toxicity of IV BCG in tumor-free mice at 60 days post-inoculation (Supplementary Fig. 3a). Our findings indicated no signs of acute toxicity induced by IV BCG treatment, including fever, weight loss (Supplementary Fig. 3b), or changes in general aspect or behavior of the animals. Liver function, as assessed by serum levels of ALT and ALP activity and albumin concentration, remained unaffected by IV BCG treatment (Supplementary Fig. 3c). Additionally, we measured serum concentrations of IL-6 and TNF as markers of possible systemic inflammatory reactions. While IL-6 was undetectable in both treated and untreated mice (Supplementary Fig. 3c), TNF was detected in both groups, with a slight increase in BCG-treated mice (Supplementary Fig. 3c). We also examined the biodistribution profile of BCG at the selected 60-day endpoint. Consistent with previous studies[23], BCG was found in the spleen, liver, lung, and mediastinal and abdominal lymph nodes (Supplementary Fig. 3d). but not in the heart or brain, indicating vaccine capacity to persist in specific organs. Indeed, this persistence may be important for protection, as lower BCG doses or heat-killed bacteria showed reduced efficacy (Fig. 1g).

Since intradermal BCG vaccination in humans has been reported to reduce lung cancer incidence[26], we aimed to evaluate the efficacy of IV BCG in a prophylactic scenario, where mice were vaccinated with IV or SC BCG 42 days prior to B16-F10 challenge. Our findings were consistent with the data observed in humans, as SC BCG vaccination

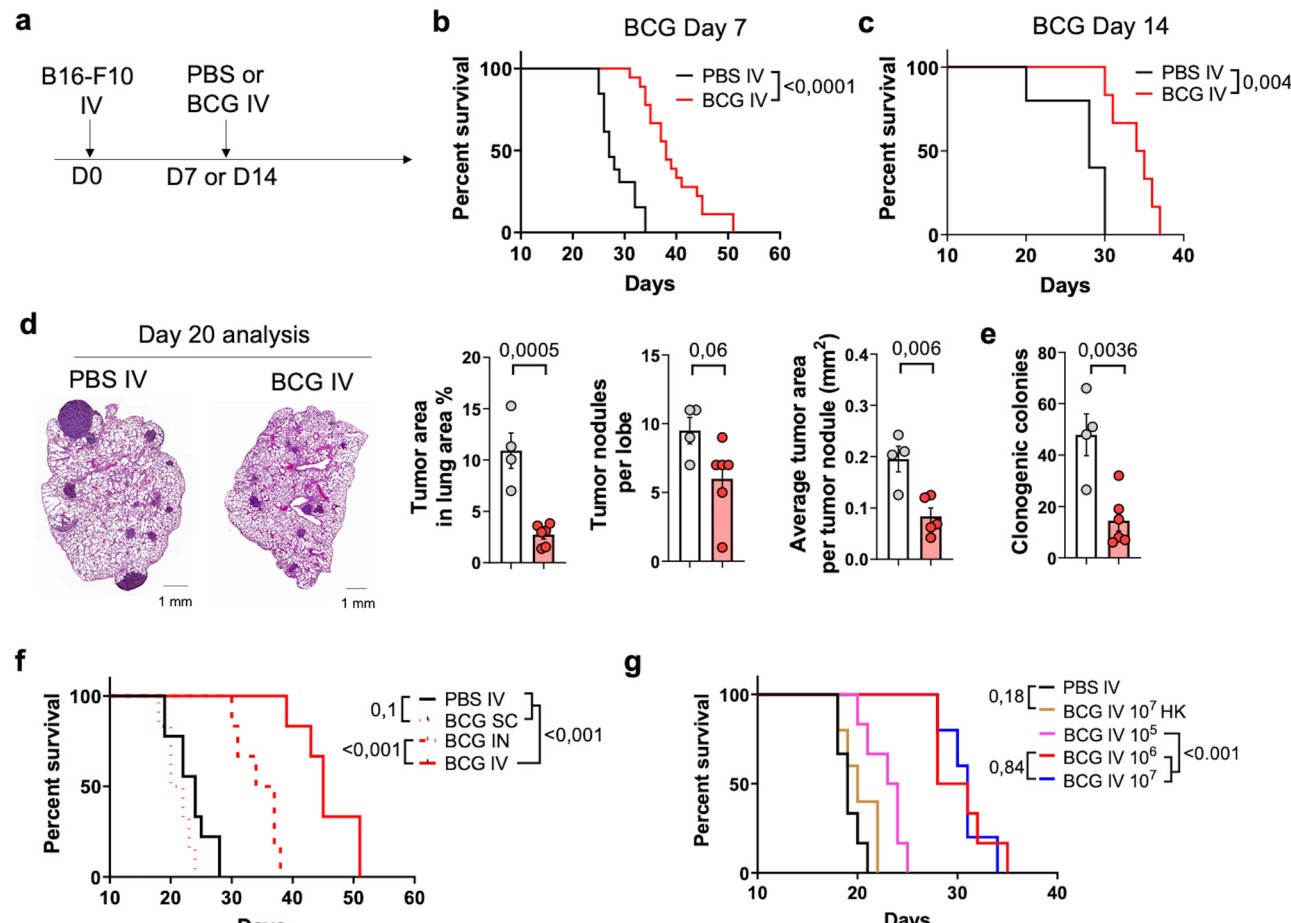

**Fig. 1 | Intravenous BCG administration delays the growth of B16-F10 lung metastases. a** Schematic diagram showing treatment strategy. **b** Survival curve of B16-F10 tumor-bearing mice treated with IV PBS ($n = 13$) or BCG ($n = 18$) at day 7, pooled from three independent experiments. **c** Survival curve of B16-F10 tumor-bearing mice treated with IV PBS ($n = 6$) or BCG ($n = 6$) at day 14, from one experiment. **d** Representative H&E images of tissue sections from tumor-bearing lungs of PBS ($n = 4$) or BCG ($n = 6$) -treated mice at day 20 after tumor cell inoculation and quantification of the tumor area, number of tumor nodules and average area per tumor nodule in lung cross-sections, representative of two independent experiments. Scale bars correspond to 1 mm in length. **e** Number of clonogenic colonies in single cell suspensions from tumor-bearing lungs shown in (**d**) ($n = 4$ mice for PBS IV and $n = 6$ for BCG IV, from one experiment. **f** Survival curve of B16-F10 tumor-bearing mice treated with IV PBS ($n = 9$), SC BCG ($n = 6$), IN BCG ($n = 6$) or IV BCG ($n = 6$), from one experiment. **g** Survival curve of B16-F10 tumor-bearing mice treated with IV PBS ($n = 6$), heat-killed IV BCG ($n = 5$) or different doses of live IV BCG ($n = 6$ mice/group), from one experiment. $P$ values were calculated using two-tailed unpaired Student's $t$ test at a 95 % CI (**d**, **e**) or log-rank (Mantel-Cox) test (**b**, **c**, **f**, **g**). Data depicted as mean ± SEM (**d**, **e**). PBS phosphate-buffered saline, IV intravenous, SC subcutaneous, IN intranasal, HK heat-killed.

resulted in a modest increase in mouse survival, from a median of 21 days to 28 (Supplementary Fig. 4a). Remarkably, prophylactic IV BCG immunization completely prevented B16-F10 tumor growth in five out of six tested mice (Supplementary Fig. 4a). This protective effect was dependent on host IFN-γ expression, as mice lacking this cytokine no longer exhibited protection (Supplementary Fig. 4b).

Finally, considering that a high proportion of the population is vaccinated with BCG (via intradermal route), we sought to determine how this pre-existing BCG-specific immunity could influence a subsequent treatment with IV BCG. To address this point, we vaccinated mice with BCG by the SC route, then challenged them with B16-F10 cells 42 days later and subsequently treated with IV BCG seven days later. Our results demonstrated that prior exposure to BCG enhanced the effectiveness of IV BCG treatment in mice with B16-F10 metastases, in comparison to BCG-naïve mice (Supplementary Fig. 4c). This finding is consistent with other studies conducted in mice and humans within the context of intravesical BCG therapy for bladder cancer[27], and provides evidence that pre-existing BCG-specific immunity does not hinder the success of this therapy, at least in our experimental scenario.

## The adaptive immune response drives IV BCG antitumoral efficacy

We next aimed to investigate the immunological pathways involved in the antitumoral effect of IV BCG treatment. Thus, we induced B16-F10 lung tumors in mice lacking either interferon gamma (*IFNγ*-/-) or perforin (*Perf*-/-) and found that the absence of these effector molecules completely abolished the therapeutic benefit conferred by IV BCG (Fig. 2a). We further depleted CD4+ and CD8+ T cells to determine their contribution. Starting the in vivo depletion process at day 10 after BCG administration, we found that both subsets of T cells were required for the therapeutic effect of the vaccine (Fig. 2b). These results indicated a central role of the adaptive immune system, dependent on IFN-γ and the cytotoxic effector molecule perforin, in mediating the antitumoral effect of IV BCG.

Next, we evaluated the effect of IV BCG on the immune landscape of B16-F10 tumor-bearing lungs at day 20, a time point where IV BCG efficacy had been demonstrated (Fig. 1d). For this, we performed an unbiased analysis of lung single-cell suspensions by spectral flow cytometry (Supplementary Fig. 5a). Our results revealed changes in both myeloid and lymphoid compartments (Supplementary Fig. 5a). IV

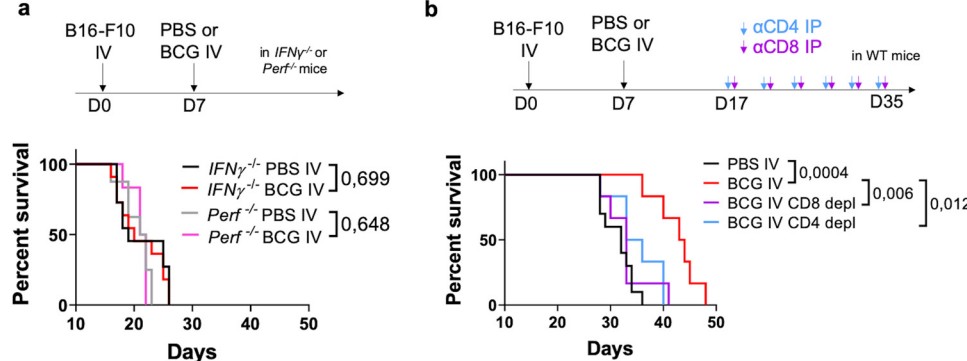

**Fig. 2 | The adaptive immune system is required for IV BCG antitumoral efficacy. a** Survival curve of B16-F10 tumor-bearing perforin ($Perf^{-/-}$) or IFN-γ deficient ($IFN\gamma^{-/-}$) mice treated with IV BCG at day 7, $n = 11$ mice for $IFN\gamma^{-/-}$ PBS IV, $n = 11$ for $IFN\gamma^{-/-}$ BCG IV, $n = 8$ for $Perf^{-/-}$ PBS IV, $n = 6$ for $Perf^{-/-}$ BCG IV, pooled from two independent experiments. **b** Survival curve of B16-F10 tumor-bearing mice receiving CD4$^+$ or CD8$^+$ T cell depleting antibodies and treated with IV BCG at day 7, $n = 6$ mice per group, from one experiment. P values were calculated using log-rank (Mantel-Cox) test (**a**, **b**). PBS phosphate-buffered saline, IV intravenous, IP intraperitoneal, WT wild-type, IFNγ interferon gamma, Perf perforin.

BCG specifically increased the proportion of monocytes, monocyte-derived Ly6C$^+$ macrophages, interstitial Ly6C$^-$ macrophages and DCs among total CD45$^+$ immune cells, while no changes in neutrophil and alveolar macrophage frequencies were observed (Supplementary Fig. 5b). Interestingly, BCG treatment strongly increased the proportion of macrophages expressing MHC-II$^+$ in both the Ly6C$^+$ and Ly6C$^-$ subsets (Supplementary Fig. 5c), suggesting an activated status of these cells. In the lymphoid compartment, IV BCG decreased B cells and increased NK cell proportions among total immune cells, while no changes were observed either in the γδ or αβ subsets of T cells (Supplementary Fig. 5d). Since T cells were found to be necessary for IV BCG efficacy in this model, we focused on this cellular subset and found that BCG treatment increased the activation status of CD4$^+$ and CD8$^+$ T cells, as evidenced by a higher proportion of CD44$^+$ CD62L$^-$ cells in both subsets (Supplementary Fig. 5e). Moreover, PD-1 expression on lung NK, γδ, or αβ T cells remained unaffected by BCG treatment (Supplementary Fig. 5f).

Next, we analyzed lung T cell function at day 20 post-tumor implantation by flow cytometry (Fig. 3a). Both CD4$^+$ and CD8$^+$ T lymphocytes from the BCG-treated group exhibited enhanced IFN-γ and IL-2 production after ex vivo polyclonal restimulation (Fig. 3b, c). Then, we performed in vivo CD45 IV labeling to discriminate between immune cells infiltrating lung tissue and immune cells in the vasculature (Fig. 3d). Tissue-infiltrating CD8$^+$ T cells from mice receiving IV BCG significantly expressed more Granzyme B than the control PBS group (Fig. 3d), demonstrating increased effector function of this cellular compartment. Regarding lung-infiltrating CD4$^+$ T cells, we examined the expression of the transcription factors T-bet and GATA3, responsible of Th1 and Th2 responses, respectively. Our results demonstrated that CD4$^+$ T cells from mice receiving IV BCG were skewed towards a Th1 profile, with a higher percentage expressing T-bet and a smaller percentage expressing GATA3 (Fig. 3e).

Next, we assessed the capacity of IV BCG to trigger lung tumor antigen-specific immune responses. To investigate this, we inoculated mice with B16-F10 cells expressing the LCMV H2-D$^b$ restricted peptide gp$_{33-41}$ (B16-F10-gp33) as a model antigen, enabling us to track tumor-specific responses (Fig. 3f). We observed a higher percentage of gp33-specific CD44$^+$ CD8$^+$ T cells in BCG-treated mice, both in lungs and spleen (Fig. 3g, h). Additionally, a greater fraction of lung gp33-specific CD8$^+$ T cells expressed Granzyme B in treated mice (Fig. 3i), suggesting improved effector function within the tumor-specific CD8$^+$ T cell population. To evaluate the functionality of tumor-specific responses, we isolated splenocytes from treated or untreated tumor-bearing mice and assessed their cytotoxicity in vitro against B16-F10 cells. IV BCG treatment enhanced the cytotoxicity exerted by splenocytes by more than two-fold compared to PBS control (Fig. 3j). This cytotoxicity was tumor-specific, as this effect was not observed when non-antigenically related LLC cells were used as targets (Fig. 3j) Furthermore, to validate our findings in vivo, we reconstituted $Rag1^{-/-}$ mice (lacking T and B cells) with splenocytes from mice bearing B16-F10 lung tumors, treated with either IV PBS or BCG, or with splenocytes from tumor-free mice immunized with IV BCG as a control. Following reconstitution, mice were challenged with B16-F10 via SC route (Fig. 3k). Our results revealed that tumors grew at a slower rate in $Rag1^{-/-}$ mice reconstituted with splenocytes from tumor-bearing mice treated with BCG compared with control groups (Fig. 3k). Taken together, these findings demonstrate that IV BCG treatment induces functional tumor-specific CD8$^+$ T cell responses against lung tumors.

## Batf3-dependent Dendritic Cells are essential for the therapeutic effect of IV BCG

We then focused on the role of lung-infiltrating conventional type 1 dendritic cells (cDC1s), given their crucial function in initiating CD8$^+$ T cell tumor-specific responses[28–30]. Analysis of tumor-bearing lungs at day 20 revealed that IV BCG treatment resulted in an increased number and frequency of lung cDC1s (Fig. 4a). Additionally, IV BCG enhanced the activation status of lung cDC1s, evidenced by the higher expression of the costimulatory molecules CD86 and CD40, as well as the XCL1 chemokine receptor XCR1 (Fig. 4b). IV BCG treatment promoted higher secretion of the Th1-inducer cytokine, IL-12, by lung cDC1s after ex vivo restimulation (Fig. 4c). We also observed a higher number of cDC1s in the lung-draining mediastinal lymph nodes (mLN) (Fig. 4d) and an enhanced surface expression of CD86, CD40, and XCR1 (Fig. 4e), suggesting that BCG treatment induced the migration of activated cDC1s from the tumor site to the mLN, where they are expected to initiate adaptive immune responses.

To determine the functional requirement of these cells for the in vivo efficacy of IV BCG, we utilized mice deficient in the transcription factor Batf3, which selectively lack cDC1s[28]. Analysis conducted at day 20 after tumor cell inoculation (Fig. 4f) revealed that IV BCG did not reduce B16-F10 lung metastases in $Batf3^{-/-}$ mice, in contrast to WT mice (Fig. 4f), which correlated with a lower infiltration of CD8$^+$ T cells in the lungs (Fig. 4g). Notably, IV BCG-induced tumor-specific response was completely abrogated in $Batf3^{-/-}$ mice, evidenced by dextramer staining (Fig. 4h) and the absence of functional cytotoxicity against B16-F10 tumor cells in vitro (Fig. 4i). Ultimately, $Batf3^{-/-}$ mice failed to benefit from the survival advantage conferred by IV BCG treatment against B16-F10 lung tumor growth (Fig. 4j). Overall, these findings demonstrate that the stimulation of tumor-specific responses in the lung by IV BCG treatment is entirely dependent on Batf3-dependent cDC1s.

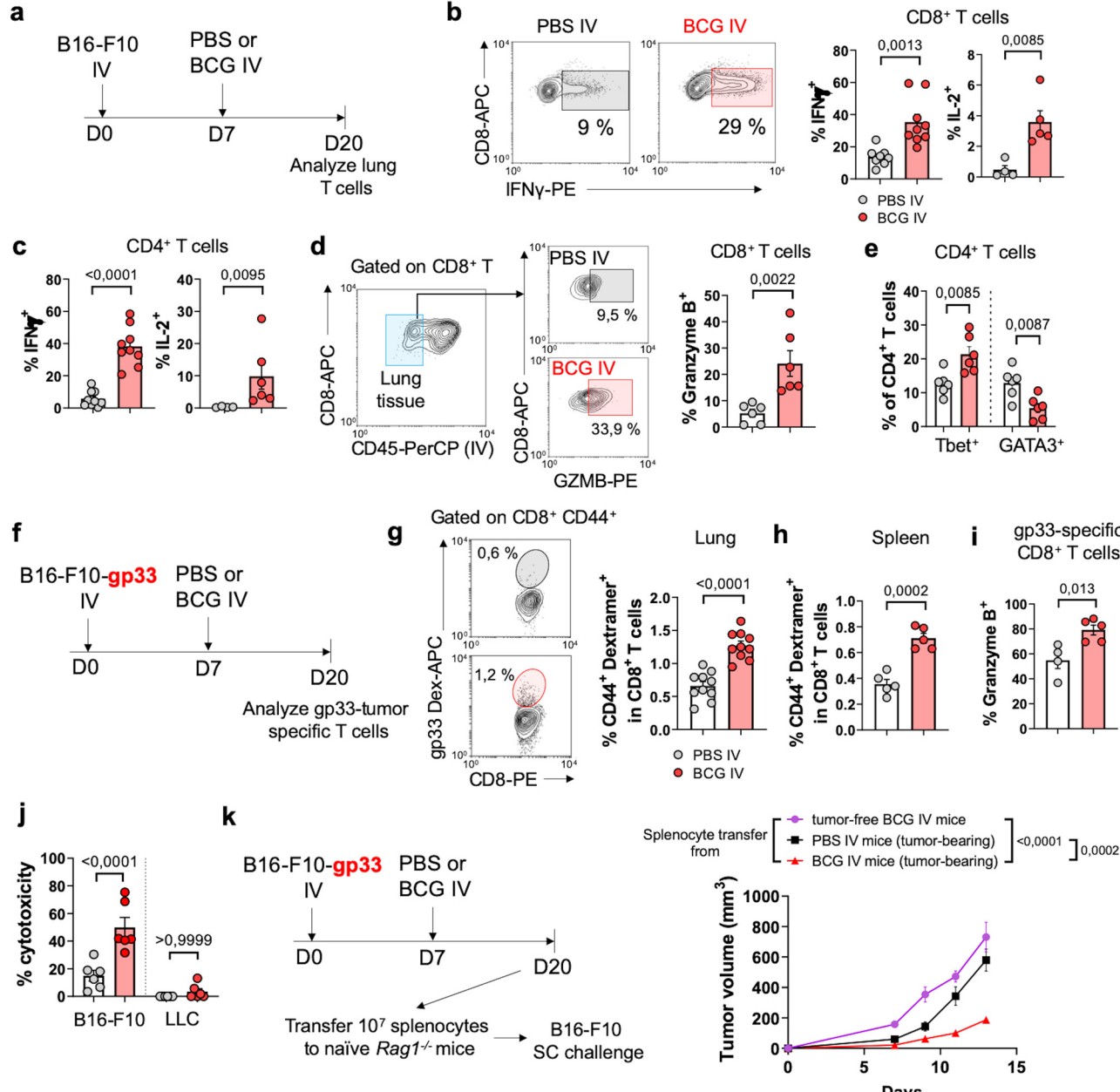

**Fig. 3 | Stimulation of T cell function in the lung by IV BCG. a** Schematic diagram showing treatment strategy for (**b**–**e**). **b, c** IFN-γ or IL-2 expression by lung CD8⁺ or CD4⁺ T cells in B16-F10 tumor bearing mice at day 20, $n = 9$ mice/group for IFN-γ and $n = 5$ for IL-2, pooled from two independent experiments. **d** Expression of Granzyme B on lung tissue-infiltrating CD8⁺ T cells; representative contour plots for the identification of lung-tissue infiltrating CD8⁺ T cells (CD45 IV⁻) and for Granzyme B expression are shown, $n = 6$ mice/group, pooled from two independent experiments. **e** Expression of T-bet and GATA3 on lung tissue-infiltrating CD4⁺ T cells, $n = 6$ mice/group, pooled from two independent experiments. **f** Schematic diagram showing treatment strategy for panels (**g**–**j**). **g, h** Quantification of CD44⁺ gp33-specific CD8⁺ T cells in the lungs and spleens of mice bearing B16-F10.gp33 lung tumors at day 20. Representative contour plots are shown. Lung (**g**): $n = 10$ mice/group, pooled from two independent experiments. Spleen (**h**): $n = 5$ mice/group,

from one experiment. **i** Granzyme B expression by gp33-specific CD8⁺ T cells in the lung, $n = 5$ mice/group, from one experiment. **j** Cytotoxicity exerted by splenocytes isolated from mice bearing B16-F10.gp33 lung tumors against target B16-F10-ZsGreenLuc or LLC-ZsGreenLuc tumor cells. Percentage cytotoxicity was calculated in reference to luminescence emitted by cells without splenocytes, $n = 6$ mice/group, from one experiment. **k** Schematic diagram showing experimental setup and follow-up of B16-F10 tumor growth in $Rag1^{-/-}$ mice reconstituted with splenocytes from the indicated donor mice, $n = 4$ BCG IV (tumor-free), $n = 6$ tumor-bearing + PBS IV, $n = 6$ tumor-bearing + BCG IV. $P$ values were calculated using two-tailed unpaired Student's $t$-test at a 95 % CI (**b**–**e**, **g**–**j**) or two-way ANOVA (**k**). Data depicted as mean ± SEM (**b**–**e**, **g**–**k**). PBS phosphate-buffered saline, IV intravenous, SC subcutaneous.

## Activation of NK cells by IV BCG

NK cells play a crucial role in controlling disseminated cells and metastasis[31]. Furthermore, BCG has been shown to activate human NK cells both in vitro[32,33] and in vivo[34,35]. Considering these precedents, we investigated whether NK cells also contributed to the antitumoral mechanism driven by IV BCG. Our findings demonstrated that IV BCG

treatment led to recruitment of NK cells to tumor-bearing lungs (Fig. 5a). Further phenotypic characterization revealed that NK cells from BCG-treated mice expressed more Granzyme B, higher levels of IFNγ and CD107a (following ex vivo stimulation) and higher surface expression of CD11b, a marker associated with activation and maturity of NK cells (Fig. 5b)[36]. Furthermore, NK cells isolated from lung cellular

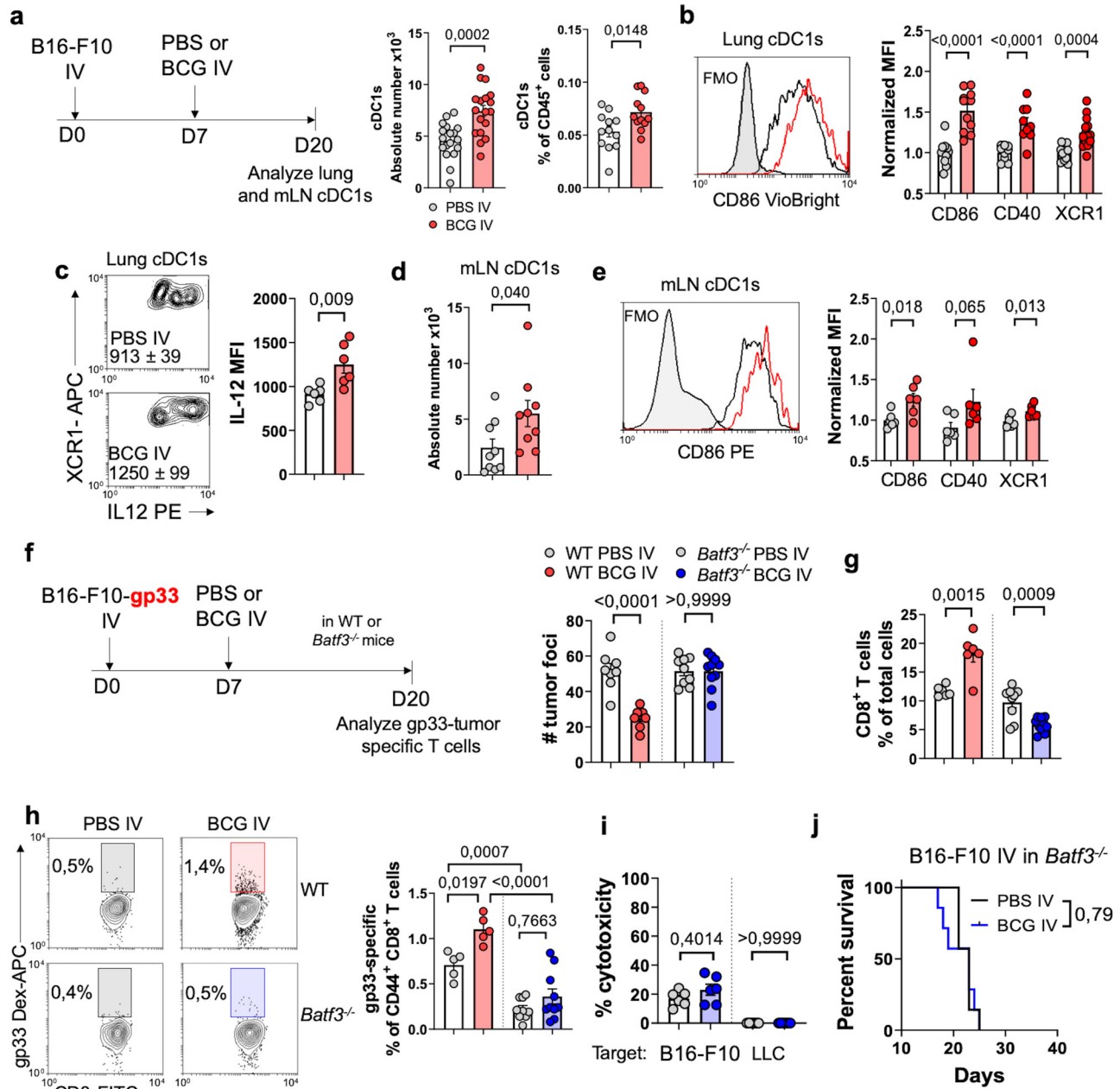

**Fig. 4 | Batf3-dependent cDC1s are required for IV BCG efficacy against B16-F10 lung metastases. a** Schematic diagram showing treatment strategy for (**a**–**e**) and absolute number and frequency of cDC1s in the lung (*n* = 18 mice per group, pooled from three independent experiments). **b** Flow cytometry quantification of CD86, CD40 and XCR1 mean fluorescence intensity (MFI) on lung cDC1s (*n* = 11 mice/group for CD86, *n* = 9 mice/group for CD40 and *n* = 14 mice/group for XCR1, pooled from four independent experiments). **c** IL-12 expression by lung cDC1s after ex vivo restimulation (*n* = 6 mice per group, from one experiment). **d** Absolute number of cDC1s in the lung-draining mediastinal lymph node (mLN) (*n* = 9 mice per group, pooled from two independent experiments). **e** CD86, CD40 and XCR1 expression by mLN cDC1s (*n* = 6 mice per group, pooled from two independent experiments). **f** Schematic diagram showing treatment strategy for (**f**–**i**) and quantification of B16-F10-gp33 lung surface metastases (*n* = 8 mice/group for WT mice and *n* = 10 mice/group for *Batf3*$^{-/-}$, pooled from two independent

experiments). **g** Lung CD8$^+$ T frequency in B16-F10-gp33 lung tumor bearing mice (*n* = 6 mice/group for WT mice and *n* = 10 mice/group for *Batf3*$^{-/-}$, pooled from two independent experiments). **h** Quantification of CD44$^+$ gp33-specific CD8$^+$ T cells in the lungs. Representative contour plots are shown. *n* = 6 mice/group for WT mice and *n* = 10 mice/group for *Batf3*$^{-/-}$ mice, pooled from two independent experiments. **i** Cytotoxicity exerted by splenocytes isolated from the indicated groups of mice bearing B16-F10-gp33 lung tumors against target B16-F10-ZsGreenLuc or LLC-ZsGreenLuc tumor cells. *n* = 6 mice/group, from one experiment. **j** Therapeutic efficacy of IV BCG in *Batf3*$^{-/-}$ mice bearing B16-F10 lung tumors (*n* = 7 mice/group, pooled from two independent experiments). *P* values were calculated using two-tailed unpaired Student's *t* test at a 95 % CI (**a**–**e**), one-way ANOVA with Bonferroni multiple-comparison test (**f**–**i**) or log-rank (Mantel-Cox) test (**j**). Data is depicted as mean ± SEM (**a**–**i**). PBS phosphate-buffered saline, IV intravenous, cDC1s type 1 conventional dendritic cells, mLN mediastinal lymph node.

suspensions of IV BCG-treated mice displayed enhanced cytotoxicity against B16-F10 cells in vitro (Fig. 5c), indicating a functional activation induced by BCG.

To specifically evaluate the ability of BCG-activated NK cells to reject tumors in vivo, we utilized a B16-F10 cell line engineered to lack

MHC-I expression by deleting the β2-microglobulin gene (Supplementary Fig. 6). This made the cells unrecognizable to CD8$^+$ T cells but vulnerable to NK cell-mediated killing (Supplementary Fig. 7a). Remarkably, mice bearing B16-F10-*B2m*$^{-/-}$ lung tumors still benefited from IV BCG treatment in a perforin-dependent manner (Fig. 5d),

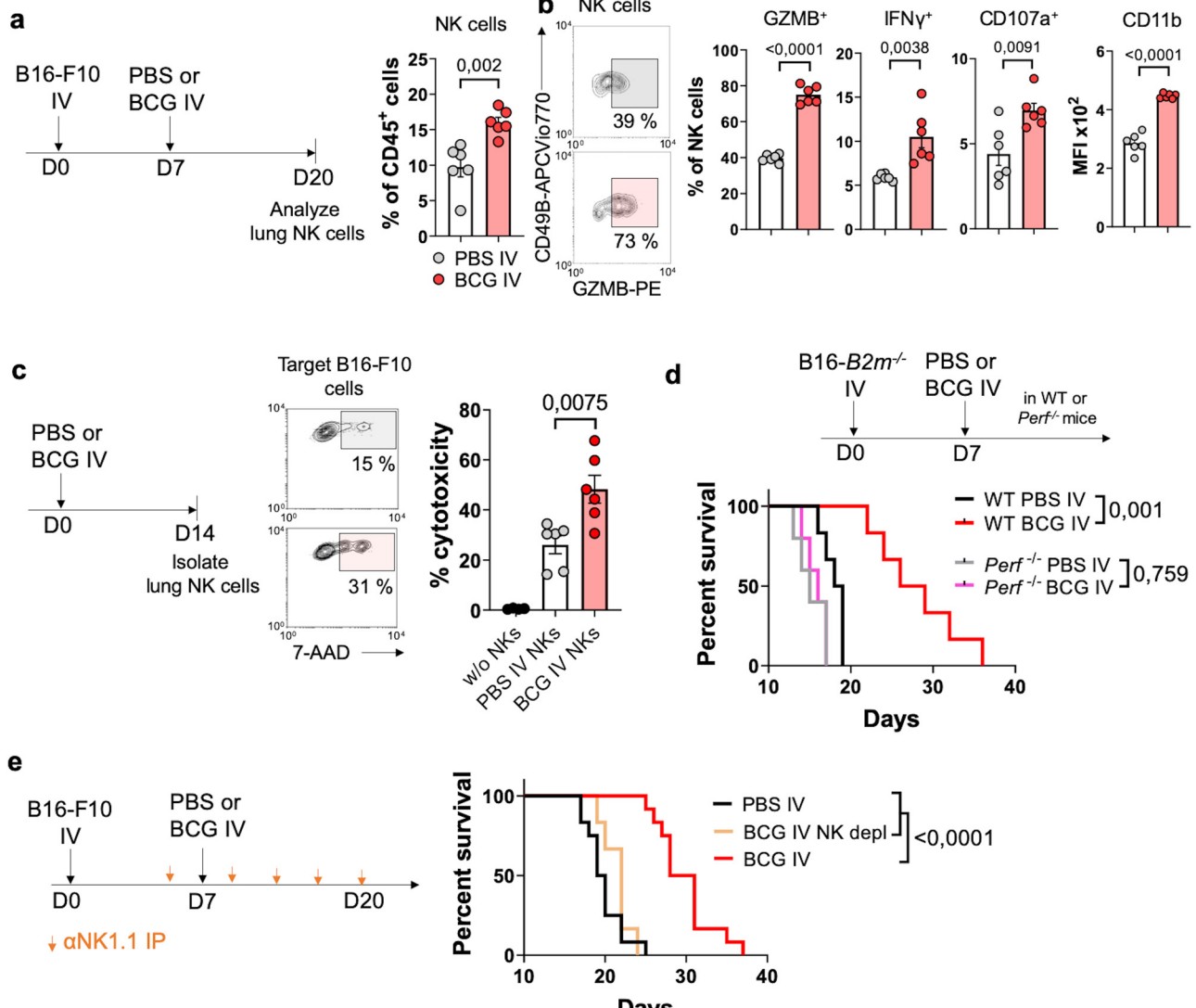

**Fig. 5 | IV BCG boosts NK cell antitumor function. a** Schematic diagram showing treatment strategy for panels (**a**, **b**) and NK cell frequency in the lungs ($n = 6$ mice/group, from one experiment representative of three independent experiments). **b** Quantification of Granzyme B, IFN-γ, CD107a and CD11b expression in lung NK cells ($n = 6$ mice/group, from one experiment representative of three independent experiments). **c** Schematic diagram showing experimental setup and in vitro cytotoxicity of purified lung NK cells against target B16-F10 tumor cells, pooled from three independent experiments with $n = 2$ mice per condition. **d** Survival

curve of WT or $Perf^{-/-}$ mice bearing B16-F10-$B2m^{-/-}$ lung metastases ($n = 6$ mice for WT and $n = 5$ for $Perf^{-/-}$ mice, from one experiment). **e** Survival curves of mice bearing B16-F10 lung metastases ($n = 12$ mice for PBS IV and BCG IV and $n = 6$ mice for BCG IV + NK depletion, pooled from two independent experiments). *P* values were calculated using two-tailed unpaired Student's *t* test at a 95 % CI (**a**–**c**) or log-rank (Mantel-Cox) test (**d**, **e**). Data is depicted as mean ± SEM. PBS phosphate-buffered saline, IV intravenous, $B2m$ Beta-2 microglobulin, Perf perforin, NK natural killer, GZMB granzyme B, IP intraperitoneal.

showing that BCG-activated NK cells can eliminate tumor cells independently from CD8+ T cell activity via perforin-mediated cytotoxicity. These results collectively support our observation that IV BCG enhances NK cell activation and cytotoxic function in the lung.

### NK cells are crucial for generation of tumor-specific T cell responses triggered by IV BCG

To assess the contribution of NK cells against MHC-I sufficient B16-F10 tumors, we removed NK cells by administering neutralizing antibodies at the time of BCG treatment (Fig. 5e). Our data demonstrated that NK cell depletion abolished the survival advantage conferred by IV BCG (Fig. 5e) Analysis at day 20 confirmed this result, as the reduction of lung metastases induced by IV BCG was lost after NK cell depletion (Fig. 6a). Notably, the depletion of NK cells in BCG-treated mice resulted in a reduction of the frequency of gp33-specific CD8+ T cells in the lungs (Fig. 6b) and diminished in vitro B16-F10-specific cytotoxicity

compared to non-depleted mice (Fig. 6c), indicating the key role of NK cells in generating tumor-specific responses driven by IV BCG. Supporting this finding, depletion of NK cells reduced IFN-γ expression by lung CD4+ and CD8+ T cells after ex vivo restimulation in BCG-treated mice (Fig. 6d).

Next, we explored the possibility that cDC1s could mediate NK cell activation in response to IV BCG, since cDC1s have been reported to recruit and stimulate NK cell function in tumors[16,37]. However, our findings did not support this hypothesis, as we observed similar activation of lung NK cells in both $Batf3^{-/-}$ and WT mice (Supplementary Fig. 7b). Moreover, IV BCG treatment effectively protected $Batf3^{-/-}$ mice inoculated with B16-F10-$B2m^{-/-}$ cells (Supplementary Fig. 7c), indicating that BCG-activated NK cells exerted in vivo cytotoxicity independently of cDC1s.

On the contrary, we found that NK cell stimulation by BCG preceded cDC1 activation. NK cells have been previously shown to

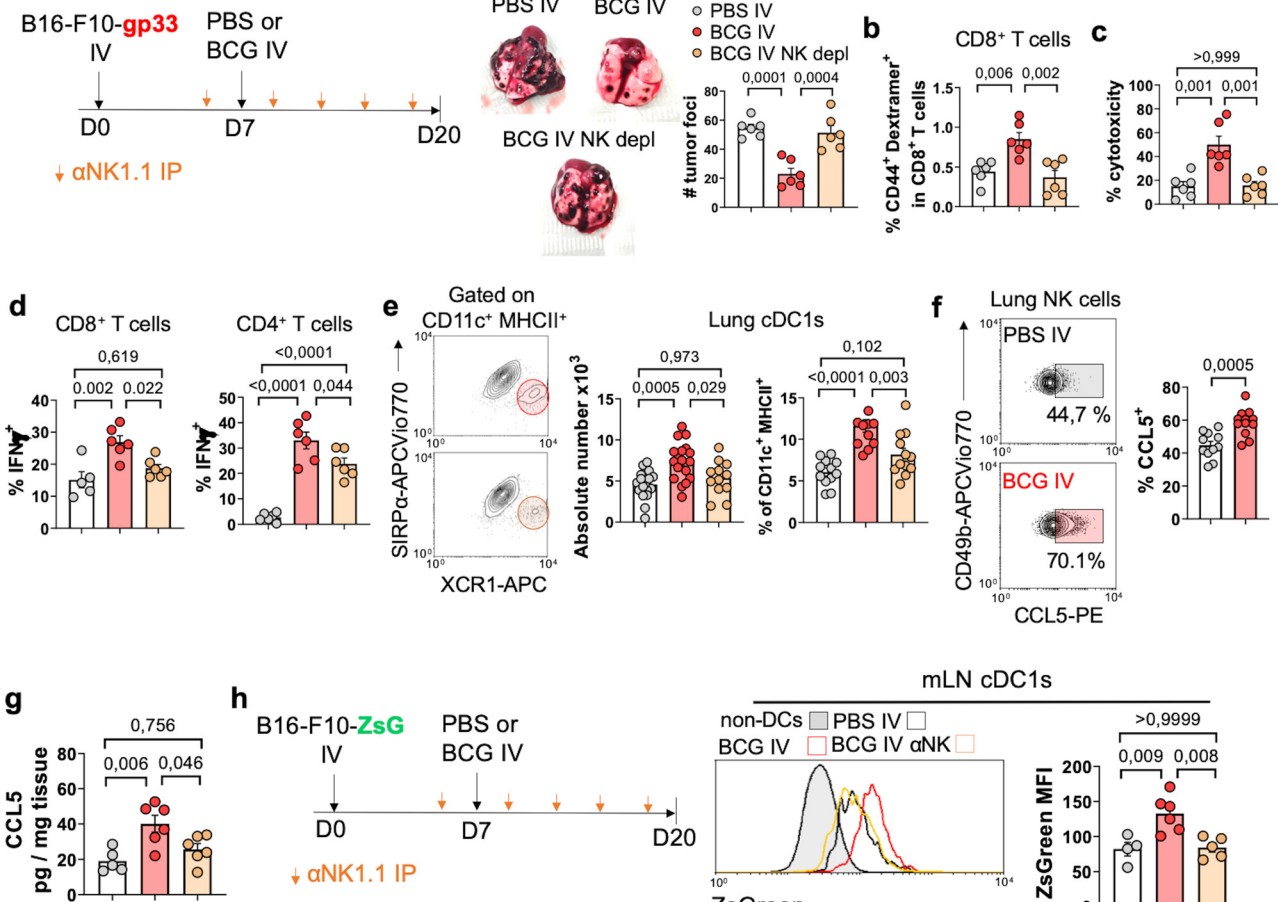

**Fig. 6 | BCG-stimulated NK cells modulate adaptive immune responses in the lung. a** Schematic diagram showing treatment strategy for (**a**–**h**) and quantification of lung surface B16-F10-gp33 metastases at day 20 (*n* = 6 mice/group, representative of two independent experiments). **b** Quantification of CD44⁺ gp33-specific CD8⁺ T cells in the lung (*n* = 6 mice/group, from one experiment). **c** Splenocytes were isolated from mice treated as in (**a**) and their cytotoxicity against B16-F10-ZsGreenLuc cells tested in an in vitro assay (*n* = 6 mice/group, from one experiment). **d** IFN-γ expression by lung CD4⁺ and CD8⁺ T cells following ex vivo restimulation (*n* = 6 mice/group, from one experiment). **e** Absolute number and frequency of cDC1s in the lungs (*n* = 18 mice/group for PBS IV and BCG IV and *n* = 12 for BCG IV + NK depletion, pooled from three independent experiments). **f** CCL5

expression by lung NK cells at day 20 (*n* = 12 mice/group, pooled from two independent experiments). **g** CCL5 protein concentration in lung homogenates (*n* = 6 mice/group, from one experiment). **h**, Schematic diagram showing treatment strategy and quantification of ZsGreen expression in mLN cDC1s (*n* = 6 mice per group, representative of two independent experiments). *P* values were calculated using two-tailed unpaired Student's *t* test at a 95 % CI (**f**) or one-way ANOVA with Bonferroni multiple-comparison test (**a**–**e**, **g**, **h**). Data is depicted as mean ± SEM. PBS phosphate-buffered saline, IV intravenous, IP intraperitoneal, NK natural killer, depl depletion, cDC1s type 1 conventional dendritic cells, mLN mediastinal lymph node, ZsG ZsGreen.

mediate infiltration of cDC1s into solid tumors[15,38]. Therefore, we analyzed whether NK cells drove to cDC1s lung recruitment in our model. We found that NK cell depletion reduced cDC1 abundance in the tumor-bearing lungs of BCG-treated mice (Fig. 6e). Previous studies demonstrated that NK cells can mediate cDC1 recruitment to tumors via secretion of CCL5[38,39]. Consistent with this finding, flow cytometry analysis revealed increased expression of CCL5 by lung NK cells following BCG treatment (Fig. 6f). Additionally, IV BCG treatment led to elevated levels of CCL5 protein in whole-lung lysates in a NK cell-dependent fashion (Fig. 6g), suggesting that NK cells are the primary source of CCL5 in response to IV BCG treatment, and therefore this chemokine may trigger cDC1 homing to lung tumors through NK cells.

Furthermore, we hypothesized that the ability of NK cells to promote tumor-specific responses might not solely rely on enhanced cDC1 recruitment, but also depend on initial killing of tumor cells to provide tumor-associated antigens to cDC1s, with the subsequent cross-priming of tumor-specific CD8⁺ T cells in the mLN. To test this hypothesis, we generated lung tumors using ZsGreen-expressing B16-F10 cells. Fluorescent protein ZsGreen maintains its

fluorescence within intracellular compartments following phagocytosis, enabling the tracking of immune cells that have engulfed tumor-associated material[40,41]. Using this approach, we observed that IV BCG treatment resulted in an increased ZsGreen signal in cDC1s from the mLN compared to untreated mice (Fig. 6h), evidencing enhanced upload of tumor-associated material. This difference in uptake was abolished when NK cells were depleted during the course of BCG treatment (Fig. 6h). Interestingly, acquisition of tumor-associated material by cDC1s was impaired in *Perf*⁻/⁻ mice. (Supplementary Fig. 8a). This finding suggests that killing of tumor cells by NK cells via granule-mediated cytotoxicity is necessary for the uptake of tumor-derived material by cDC1s. Consistent with this observation, IV BCG did not enhance tumor-specific CD8⁺ T cell responses in mice lacking perforin (Supplementary Fig. 8b), and splenocytes did not display enhanced cytotoxicity against B16-F10 cells in vitro (Supplementary Fig. 8c). Consequently, IV BCG failed to effectively reduce tumor burden in *Perf*⁻/⁻ mice (Supplementary Fig. 8d), consistent with the lack of protection observed in survival experiments (Fig. 2a).

### IV BCG immunotherapy in orthotopic lung cancer models

We next explored whether IV BCG treatment could additionally be harnessed in the setting of NSCLC. To investigate this scenario, we utilized an orthotopic model based on IV inoculation of Lewis Lung Carcinoma (LLC) cells, a $Kras^{G12C}/p53^{mut}$ cell line commonly used to study NSCLC[42–44] (Fig. 7a and Supplementary Fig. 9). At day 7 after tumor cell inoculation, histological analysis confirmed the presence of LLC tumor nodules (Supplementary Fig. 9). Delivery of IV BCG at this timepoint significantly improved mouse survival compared to the control PBS group (Fig. 7b). Another cohort of mice was analyzed at day 20, finding that IV BCG treatment at day 7 reduced tumor burden

compared to untreated mice (Fig. 7c). In addition, we evaluated the efficacy of IV BCG administered 14 days after LLC cells inoculation. At this selected time point, tumor nodules covered up ~10 % of the total lung area and were already visible to the naked eye (Supplementary Fig. 9a). Even in this higher tumor burden scenario, BCG still conferred a therapeutic benefit, increasing median survival from 19 to 25 days compared to untreated mice (Fig. 7d). Interestingly, as shown with B16-F10 tumors, administration of IV BCG before tumor challenge completely prevented LLC lung tumor growth in 60 % of the evaluated mice (Supplementary Fig. 10a). To extend our findings, we tested IV BCG in another orthotopic NSCLC model using the TC-1 cell line[45,46]. As

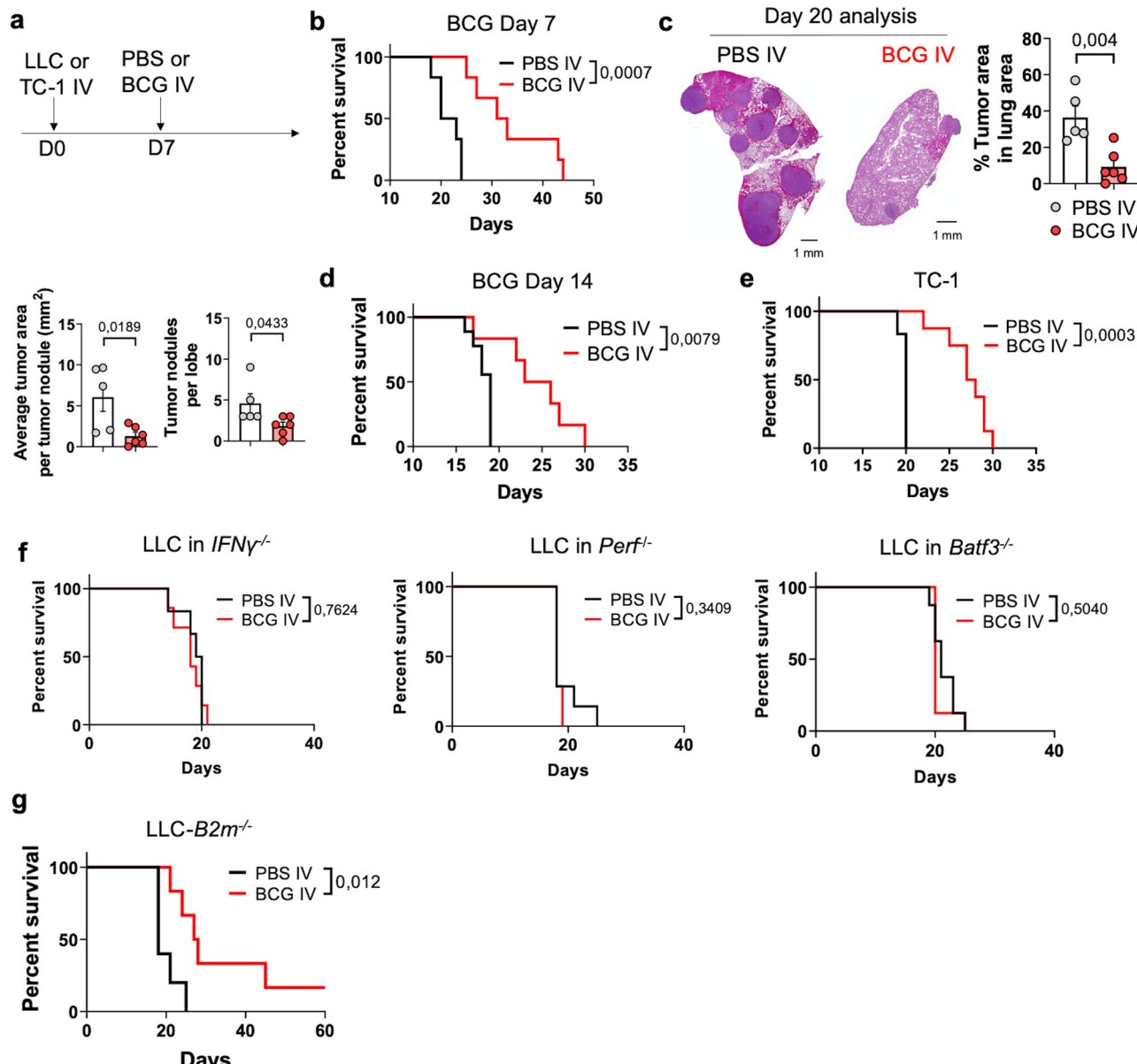

**Fig. 7 | Efficacy of IV BCG in orthotopic lung cancer models. a** Schematic diagram showing treatment strategy. **b** Survival curves of mice bearing orthotopic LLC tumors and treated with IV PBS ($n = 6$) or BCG ($n = 6$) at day 7, from one experiment representative of two independent experiments. **c** Representative H&E images of tumor-bearing lungs of PBS ($n = 5$) or BCG ($n = 6$) -treated mice (at day 7) at day 20 after inoculation of LLC tumor cells, and quantification of the tumor area, number of tumor nodules and average area per tumor nodule in lung cross-sections, from one experiment. Scale bars correspond to 1 mm in length. **d** Survival curve of mice bearing orthotopic LLC tumors and treated with IV PBS ($n = 9$) or BCG ($n = 6$) at day 14, from one experiment. **e** Survival curve of mice bearing orthotopic TC-1 tumors

and treated with IV PBS ($n = 6$) or BCG ($n = 8$) at day 7, from one experiment. **f** Survival curves of $IFN\gamma^{-/-}$, $Perf^{-/-}$ or $Batf3^{-/-}$ mice bearing orthotopic LLC tumors and treated with IV PBS or BCG at day 7 ($n = 8$ mice/group, pooled from two independent experiments). **g** Survival curve of mice bearing orthotopic LLC-$B2m^{-/-}$ tumors and treated with IV PBS ($n = 6$) or BCG ($n = 6$) at day 7, from one experiment. $P$ values were calculated using two-tailed unpaired Student's $t$ test at a 95 % CI (**c**) or log-rank (Mantel-Cox) test (**b**, **d**–**g**). Data depicted as mean ± SEM (**c**). PBS phosphate-buffered saline, IV intravenous, NK natural killer, Perf perforin, $B2m$ Beta-2 microglobulin.

expected, histological analysis at day 7 following tumor inoculation revealed the presence of lung tumor nodules, which exhibited rapid growth in untreated mice (Supplementary Fig. 10b). In this model, BCG treatment at day 7 significantly prolonged mice survival from a median of 20 to 28 days compared to control PBS-treated mice (Fig. 7e).

We then aimed to characterize the immune response against orthotopic lung tumors. Similar to the B16-F10 model, the therapeutic

benefit provided by BCG in mice bearing orthotopic LLC tumors was entirely dependent on the host's expression of IFN-γ and perforin and also required *Batf3*-dependent cDC1s (Fig. 7f). Moreover, IV BCG was effective against LLC-*B2m*−/− lung tumors (Fig. 7g), suggesting that BCG-activated NK cells may also play a role in this model. Flow cytometry analysis of LLC-tumor-bearing lungs at day 20 (Fig. 8a) revealed an increased proportion of CD8+ T cells, CD4+ T cells and NK cells

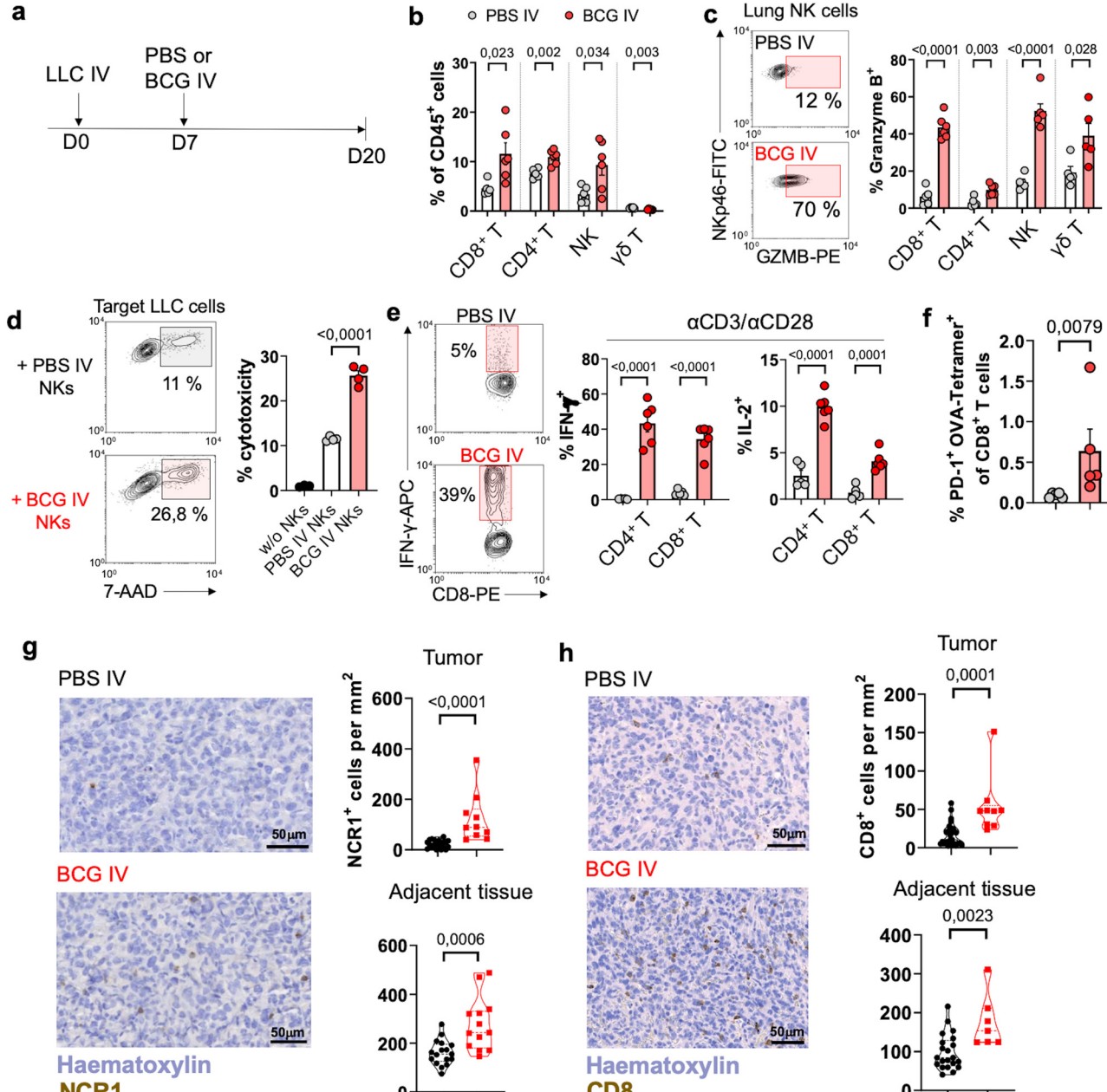

**Fig. 8 | IV BCG recruits and activates NK and CD8+ T cells in an orthotopic lung cancer model. a** Schematic diagram showing treatment strategy. **b** Frequencies of cellular subsets in the lung (*n* = 5 mice for PBS IV and *n* = 6 mice for BCG IV, from one experiment). **c** Granzyme B expression by immune cell subsets (*n* = 5 mice for PBS IV and *n* = 6 mice for BCG IV, from one experiment). **d** In vitro cytotoxicity of lung NK cells against LLC tumor cells. Pooled from two independent experiments with *n* = 2 mice per condition in each experiment. **e** IFN-γ or IL-2 expression by lung CD8+ or CD4+ T cells (*n* = 5 mice for PBS IV and *n* = 6 mice for BCG IV, from one experiment). **f** Frequency of PD-1+ SIINFEKL-specific CD8+ T cells in the lungs of mice bearing orthotopic LLC-OVA tumors (*n* = 5 mice/group, from one experiment). **g**, Representative IHC staining of NK cells in orthotopic LLC lung tumors.

NCR1+ cells were quantified inside tumor nodules (*n* = 21 regions for IV PBS-treated mice and *n* = 10 regions for IV BCG-treated mice) and adjacent tumor-free tissue (*n* = 17 regions for IV PBS and *n* = 13 regions for IV BCG-treated mice). Data comes from one experiment with *n* = 5 mice for PBS IV and *n* = 6 mice for BCG IV. **h** CD8+ T cell quantification by IHC in LLC tumor nodules (*n* = 27 regions for IV PBS-treated mice and *n* = 10 regions for IV BCG-treated mice) and adjacent tumor-free tissue (*n* = 21 regions for IV PBS and *n* = 7 regions for IV BCG-treated mice). Data comes from one experiment with *n* = 5 mice for PBS IV and *n* = 6 mice for BCG IV. Scale bars correspond to 50 μm in length (**g**, **h**). *P* values were calculated using two-tailed unpaired Student's t-test at a 95 % CI (**b**–**h**). Data depicted as mean ± SEM. PBS phosphate-buffered saline, IV intravenous, NK natural killer, GZMB Granzyme B.

following IV BCG treatment (Fig. 8b), accompanied by enhanced cytotoxic potential evidenced by increased Granzyme B expression (Fig. 8c). Consistent with our findings in the B16-F10 model, IV immunization improved the cytotoxic activity of NK cells isolated from the lung (Fig. 8d) or spleen (Supplementary Fig. 10c) in killing assays against LLC tumor cells in vitro. To validate our findings using human NK cells, we assessed the capacity of BCG to enhance NK cell cytotoxicity against human lung cancer cell lines in vitro. Peripheral blood mononuclear cells (PBMCs) from healthy donors were stimulated in vitro with BCG for 7 days, following a protocol previously reported to activate NK cells (Supplementary Fig. 10d)[32,33]. The killing of two human lung cancer target cells (H2228 and H3122) significantly increased when using BCG-activated PBMCs compared to unstimulated PBMCs (Supplementary Fig. 10d). This enhanced cytotoxicity concurred with enhanced NK cell degranulation, measured by CD107a expression following culture with target cells (Supplementary Fig. 10e).

Furthermore, lung CD4$^+$ and CD8$^+$ T cells from BCG-treated mice exhibited improved functionality, characterized by higher expression of IFN-γ and IL-2 upon ex vivo restimulation (Fig. 8e). Next, to track tumor-specific responses, we orthotopically implanted ovalbumin (OVA)-expressing LLC tumor cells. IV BCG immunization significantly increased the frequency of OVA-specific CD8$^+$ T cells in the lungs at day 20 (Fig. 8f). Similar to parental LLC tumors, a single dose of IV BCG given at day 7 inhibited the growth of LLC-OVA orthotopic tumors, resulting in apparent remission in two out of six mice (Supplementary Fig. 10f). Immune-driven tumor elimination was evident as the two mice that rejected LLC-OVA lung tumors showed resistance to subsequent subcutaneous rechallenge with parental LLC cells lacking OVA, indicating the generation of immune memory against other tumor antigens apart from OVA in these mice (Supplementary Fig. 10g).

Since the flow cytometry approach used in this study is based on whole lung homogenization, it does not allow discrimination between healthy and tumoral tissue. Therefore, we quantified CD8$^+$ and NK cells through IHC in tissue sections of mice bearing orthotopic LLC tumors to determine whether IV BCG induced the infiltration of these immune cell subsets into the tumor bed. We observed that IV BCG increased the abundance of NCR1$^+$ NK cells (Fig. 8g) and CD8$^+$ T lymphocytes (Fig. 8h) both inside the tumor and at the adjacent tissue. Altogether, these findings show that IV BCG not only enhances the cytotoxic potential of lung NK and CD8$^+$ T cells but also facilitates their recruitment into orthotopic LLC lung tumors, resulting in slower tumor growth and enhanced survival.

Additionally, we evaluated the tolerability of IV BCG in LLC tumor-bearing mice at day 20 following tumor implantation (day 13 post- BCG administration) (Supplementary Fig. 11a). Similar to the tumor-free scenario, IV BCG did not induce fever, weight loss (Supplementary Fig. 11b), or liver function alterations (Supplementary Fig. 11c) in this experimental setting. Interestingly, orthotopic LLC tumors strongly increased serum IL-6 levels, which were drastically reduced by IV BCG treatment (Supplementary Fig. 11d), coinciding with the lower tumor burden observed in these mice (Fig. 7c). This is an interesting finding since systemic induction of IL-6 in lung cancer patients correlates with worse prognosis[47], and IL -6 has been described to promote tumor growth and evasion from immunosurveillance[48,49]. Finally, TNF serum levels were slightly increased in the IV BCG group (Supplementary Fig. 11d), as observed in tumor-free mice, which could be attributed to immune system activation by the vaccine.

### IV BCG immunotherapy boosts PD-L1 checkpoint blockade efficacy

We then sought to explore whether IV BCG immunotherapy could enhance the effectiveness of immune checkpoint blockade (ICB) mediated by anti-PD-L1. IV BCG treatment increased PD-L1 surface expression on different lung myeloid cell subsets including tumor-associated macrophages (TAMs), neutrophils and DC subsets (Fig. 9a, b), but apparently not on tumor cells (Fig. 9d). This phenotype was entirely dependent on host IFN-γ (Fig. 9c), suggesting that PD-L1 upregulation on lung immune cells is a consequence of the BCG-triggered inflammatory response. Our findings demonstrated that the combination of IV BCG and ICB improved the survival of mice bearing lung B16-F10 metastases compared to IV BCG alone, whereas mono-therapy with αPD-L1 did not confer any survival advantage in this context (Fig. 9e). Analysis of the tumor-bearing lungs at day 26 revealed that the combination of IV BCG with ICB further enhanced the functionality of lung CD8$^+$ T cells, including the tumor-specific (gp33-dextramer-positive), when compared to BCG alone (Fig. 9f). IV BCG + ICB also increased the frequency of degranulating (CD107a$^+$) and IFN-γ secreting NK cells (Fig. 9f), indicating that this cellular subset was also targeted by the combinatorial approach. Consistent with these findings, mice receiving IV BCG + ICB exhibited enhanced systemic tumor-specific cytotoxic responses specifically targeting B16-F10 tumor cells (Fig. 9g), which overall correlated with the therapeutic activity observed in this model.

In the LLC orthotopic model, PD-L1 blockade alone had minimal impact on mouse survival compared to the untreated group, whereas the combination of IV BCG and ICB exhibited a complementary effect (Fig. 9h). Our data revealed that the combination of IV BCG with PD-L1 blockade was well-tolerated without evident signs of toxicity in this model (Supplementary Fig. 12). Strikingly, 7 out of the 12 mice treated with IV BCG and ICB showed apparent tumor remission at the end of the 90-day follow-up, as confirmed by histopathological analysis (Supplementary Fig. 13). These survivor mice were rechallenged subcutaneously with LLC and B16-F10 cells in opposite flanks. Our results showed complete rejection of the LLC rechallenge, while non-antigen related B16-F10 tumors grew to a similar extent as in naïve mice, indicating the induction of a long-term immunological memory response that would be functional preventing tumor relapse (Fig. 9i).

## Discussion

Here, we describe in distinct murine lung tumor models a cancer immunotherapy approach based on IV administration of the live tuberculosis vaccine BCG, whose antitumoral efficacy in NMIBC patients has been widely contrasted when instilled directly into the bladder. In this study, we demonstrate that BCG delivered intravenously induces a coordinated crosstalk among multiple immune cell populations, including T cells, NK cells, and cDC1s, leading to the generation of effective tumor-specific immune responses in the lung.

We have identified NK cells as a key cellular subset driving the therapeutic activity of IV BCG. Previous studies have described the privileged role of NK cells in orchestrating antitumor adaptive immune responses by modulating T cell functionality or by recruiting cDC1s to the tumor niche[38,39,50,51]. In our model, NK cells contributed to BCG treatment success in two ways: (1) recruiting cDC1s to the tumor-bearing lung, which could be mediated by CCL5 produced by activated NK cells; (2) killing of tumor cells by BCG-activated NK cells in a perforin-dependent way, which critically generates tumor cell-derived material that is subsequently acquired by cDC1s to trigger tumor-specific T cell responses. Importantly, data from cancer patients indicate that NK cell frequencies correlate with BDCA3$^+$ DCs, and the presence of both cellular subsets coincides with improved responses to αPD-1 immunotherapy and overall higher survival[15]. These data suggest that targeting the NK cell-cDC1 axis, as we have shown with IV BCG therapy, may hold great promise in the context of NSCLC. Furthermore, an interesting aspect of enhancing NK cell antitumor activity is the possibility to overcome T cell- associated immune evasion. Tumor cells can become resistant to CD8$^+$ T cells by selection of clones lacking MHC-I expression or with defects in IFN-sensing pathways,

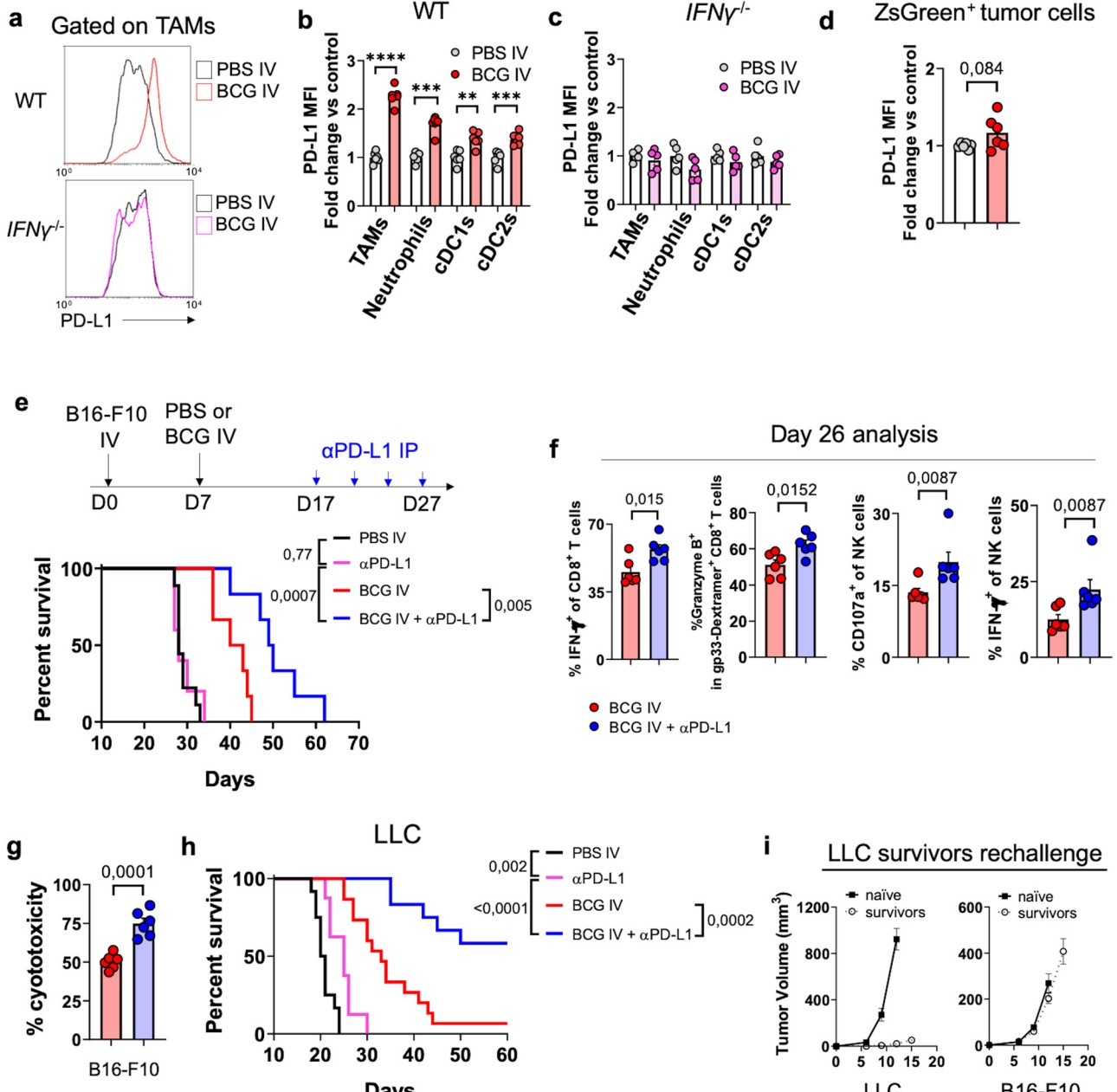

**Fig. 9 | PD-L1 checkpoint blockade boosts IV BCG efficacy in lung tumors.**
**a–c** Quantification of PD-L1 expression at day 20 in different subsets of lung immune cells from WT (**b**) or *IFNγ⁻/⁻* mice (**c**) bearing B16-F10 lung tumors treated or not with BCG IV. Representative flow cytometry histogram plots are shown (**a**), n = 5 (*IFNγ⁻/⁻*) or n = 6 (WT) mice per group, from one experiment. **d** PD-L1 expression on ZsGreen⁺ B16-F10 tumor cells in vivo at day 20, n = 6 mice/group, from one experiment. **e** Survival curves of mice bearing B16-F10 lung metastases and treated with IV PBS (n = 9), IV PBS + αPD-L1 antibody (n = 6), IV BCG (n = 6) or IV BCG + αPD-L1 antibody (n = 6), from one experiment. **f** Flow cytometry analysis of IFN-γ expression by CD8⁺ T cells, Granzyme B expression by gp33-specific CD8⁺ T cells, and CD107a and IFN-γ expression by NK cells in the lungs of mice bearing B16-F10-gp33 lung metastases at day 26 post tumor implantation and after three doses of αPD-L1 antibody at days 17, 20 and 24 (n = 6 mice/group, from one

experiment). **g** In vitro splenocyte cytotoxicity against B16-F10-ZsGreenLuc cells (n = 6 mice/group, from one experiment). **h** Survival curves of mice bearing orthotopic LLC lung tumors and treated with IV PBS (n = 12), IV PBS + αPD-L1 antibody (n = 8), IV BCG (n = 15) or IV BCG + αPD-L1 antibody (n = 12) as in (**e**), pooled from two independent experiments. **i** LLC survivors or naïve mice were inoculated subcutaneously with LLC tumor cells in one flank and B16-F10 in the contralateral flank, and tumor growth was measured until the predefined endpoint (n = 6 mice in the naïve group and n = 7 mice in the rechallenge group, from two independent experiments). *P* values were calculated using two-tailed unpaired Student's *t* test at a 95 % CI (**b–d**, **f**, **g**) or log-rank (Mantel-Cox) test (**e**, **h**). Data depicted as mean ± SEM (**b–d**, **f**, **g**, **i**). PBS phosphate-buffered saline, IV intravenous, IP intraperitoneal, NK natural killer, TAMs tumor-associated macrophages, PD-L1 programmed cell death-ligand 1, cDC1s type 1 conventional dendritic cells.

which in turn increases their sensitivity to NK cell targeting[52]. In our model, MHC-I deficient lung tumors still responded to IV BCG, highlighting that therapies that engage both T and NK cells are a promising strategy to overcome tumor immune escape. However, the precise mechanism by which NK cells become activated in the lung by IV BCG remains to be studied.

Tumor resistance to ICIs has been attributed to several potential mechanisms, such CD8⁺ T cell exclusion from the tumor[12], a weak preexisting or dysfunctional CD8⁺ T cell antitumor response[42], a dysfunctional state or lack of infiltration into the tumor of cytotoxic immune cells and cDC1s[38,53,54] or an immune-suppressive TME driven by myeloid cells[55]. Our results suggest that IV BCG could overcome

several of these mechanisms, which might explain the strong anti-tumoral effect observed with the combination of IV BCG and αPD-L1 in the two different lung tumor models tested. LLC lung tumors responded poorly to PD-L1 blockade as monotherapy, which correlated with the scarce immune infiltrate found inside the tumor mass. However, IV BCG facilitated the infiltration of CD8[+] and NK cells into the tumor, and lung cytotoxic cells from BCG-treated mice showed an increased effector phenotype with higher cytotoxic potential and IFN-γ secretion. In addition, our results revealed an IFNγ-mediated increment of PD-L1 expression in different subsets of myeloid cells induced by BCG. Therefore, we speculate that this global increment of PD-L1 driven by BCG could be attributed to successful infiltration of activated IFN-γ-secreting immune cells into the tumor. Importantly, different studies have demonstrated that PD-L1 expression on myeloid cells can be a consequence of an inflamed TME and is associated with better prognosis and response to ICIs[56–60].

Lastly, our results showed that the strongest antitumoral effect of BCG was achieved by IV administration, posing an important safety concern for further clinical development. In this study, a single dose of IV BCG resulted well-tolerated in both tumor-free and tumor-bearing mice, alone and in combination with PD-L1 blocking antibodies, and did not induce signs of acute or chronic toxicity. These results agree with published studies in mice and NHP models[21,24], and even in humans[61], where IV BCG has been tested under different regimens. Furthermore, phase I clinical trials based on intravenous inoculation of other live microorganisms such as attenuated strains of *Listeria*[62,63] and *Salmonella typhimurium*[64] or oncolytic viruses[65] have been carried out in past years. Altogether, the strong antitumoral activity driven by IV BCG in mouse tumor models and the lack of apparent toxicity support further exploration of this therapeutic strategy for the treatment of lung tumors.

## Methods

### Ethics statement
Experimental work was conducted in agreement with European and National directives for protection of experimental animals, and experimental procedures were approved by the Ethics Committee for Animal Experiments of University of Zaragoza (PI46/18, PI33/15, and PI50/14).

### Mouse strains
Male and female mice between the ages of 8 and 12 weeks were used. C57BL/6JR mice were purchased to Janvier Biolabs. Mouse strains deficient for interferon gamma (*IFNγ*[-/-], strain #002287) and Rag1 (*Rag1*[-/-], strain #002216) bred on C57BL/6JR background were purchased from Jackson Laboratories. The mouse strains deficient for Perforin (*Perf*[-/-])[66] and Batf3 (*Batf3*[-/-])[67] on C57BL/6JR background were bred in the facilities of the Centro de Investigaciones Biomédicas de Aragón (CIBA). Mouse experimentation and breeding were done in a SPF-facility at 20−24 °C, 50−70% humidity, and a light-dark cycle of 12 h.

### Tumor cell lines
B16-F10 and B16-F10-gp33[68] cells were given by Dr. Julián Pardo (University of Zaragoza). LLC cells were given by Dr. David Sancho (Centro Nacional de Investigaciones Cardiovasculares, Madrid). LLC-OVA tumor cells were kindly provided by Dmitry Gabrilovich (University of Pennsylvania). TC-1 cells were given by Dr. T.C. Wu (Johns Hopkins Medicine)[69]. B16-F10-ZsGreenLuc and LLC-ZsGreenLuc cells were made in the laboratory by transfection with a lentivirus encoding ZsGreen and firefly luciferase Luc2P (pHIV-Luc2-ZsGreen, from Addgene) and sorted based on high ZsGreen expression. For the generation of LLC-*B2m*[-/-] and B16-F10-*B2m*[-/-] cell lines, parental cells were transfected with CRISPR-Cas9 plasmids targeting the β2-microglobulin gene (purchased from SantaCruz Biotechnology) and

cells were selected with puromycin and then sorted based on lack of MHC-I expression after staining with an antibody directed to H2K[b]/D[b] (Miltenyi). Tumor cells were cultured with complete DMEM, containing 10% inactivated Fetal Bovine Serum (FBS, Gibco), Glutamax (Gibco) and penicillin/streptomycin (Gibco) and were always used with less than 8 passages from thawing. The B16-F10-gp33 and LLC-OVA clones were maintained in complete DMEM containing 500 µg ml[-1] of G418 (Gibco).

### Tumor outgrowth studies
For lung tumor induction, $5 \times 10^4$ B16-F10 or B16-F10-gp33, $1 \times 10^5$ B16-*B2m*[-/-], $1,5 \times 10^5$ B16-F10-ZsGreenLuc, and $4 \times 10^5$ LLC, LLC-*B2m*[-/-] or LLC-OVA cells were injected intravenously in serum-free DMEM. Mice were sacrificed by inhaled anesthesia followed by cervical dislocation based on a clinical score including evaluation of weight loss (a maximum of 20% of initial weight loss was allowed), general appearance and activity, and the frequency and quality of breathing, which was performed every 2 days for the duration of the experiment.

For subcutaneous tumors, $1 \times 10^6$ LLC and $1 \times 10^6$ B16-F10 cells resuspended in serum-free DMEM were injected in contralateral flanks. Size of subcutaneous tumors was measured with a digital caliper three times per week and determined by using the following formula: [(tumor width)$^2 \times$ (tumor length)]/2. Mice were sacrificed by inhaled anesthesia followed by cervical dislocation either when tumor volume exceeded 1 cm$^3$, when tumor diameter exceeded 10 cm in any direction, or when tumors became ulcerated.

### Human tumor cell lines and PBMCs
The human lung cancer cell lines H1322 (bronchi-alveolar carcinoma) and H2188 (lung adenocarcinoma) were obtained from Dr. A. Romero (Puerta de Hierro Hospital, Madrid) and authenticated by satellite genotyping at the genomics service of the Instituto de Investigaciones Biomédicas (IIB-CSIC).

PBMCs from buffy coats of healthy donors were obtained from the Regional Transfusion Centre (Madrid) with ethical permission and experimental protocols approved by the institutional committees: Regional Transfusion Centre (PO-DIS-09) and assessed by the bioethics committee of CSIC. Informed consent was obtained at the Transfusion Centre from all participants. All methods were carried out in accordance with biosafety guidelines and regulations authorized by CNB-CSIC.

PBMCs were isolated by centrifugation on Ficoll-HyPaque and cultured in RPMI-1640 (Biowest) supplemented with 5% FBS, 5% human male AB serum, 2 mM glutamine, 1 mM sodium pyruvate, 0.1 mM non-essential amino acids, 10 mM Hepes, 100 U/ml penicillin and 100 U/ml streptomycin (Biowest). For BCG stimulation, experiments were performed as previously described (Garcia-Cuesta et al., 2017). Briefly, $0.5 \times 10^6$ PBMCs/ml were incubated in 48-well plates with or without BCG Pasteur at a 6:1 ratio (total bacteria to PBMC). One week later, cells in suspension were recovered from the co-culture, centrifuged, and analyzed by flow cytometry.

### Bacterial strains
The BCG Pasteur 1173P2 (a kind gift from Dr. Roland Brosch, Institute Pasteur, France) used in this study was grown at 37 °C in Middlebrook 7H9 broth (BD Difco) supplemented with 0.05% Tween 80 (Sigma) and 10% Middlebrook albumin dextrose catalase enrichment (ADC; BD Biosciences), or on solid Middlebrook 7H10 agar (BD Difco) supplemented with 10% ADC (BD Biosciences). The GFP-expressing BCG strain was generated in the laboratory by transformation with the pJKD6 plasmid (a kind gift from Luciana Leite, Butantan Institute, Brazil). A solution containing $10^6$ CFUs (or $10^5$ or $10^7$ when indicated) of BCG Pasteur diluted in PBS was inoculated either intravenously, subcutaneously or intranasally in mice anesthetized with isoflurane.

## Antibody-based cell depletion and treatments

For CD4 and CD8 depletion, mice were injected intraperitoneally with 200 μg of anti-CD4 (clone GK1.5, BioXCell) or 200 μg of anti-CD8α (clone 2.43, BioXCell), and repeated doses were administered to achieve continuous depletion. For antibody-based treatments, 200 μg of anti-PD-L1 (clone 10 F.9G2, BioXCell) was injected intraperitoneally twice a week for a total of four doses. NK cells were depleted with repeated doses of 100 μg of αNK1.1 (clone PK136, BioXCell).

## Preparation of single-cell suspensions

Lungs were aseptically removed and homogenized in DMEM containing deoxyribonuclease I (DNase I, 40 U ml$^{-1}$; AppliChem) and collagenase D (2 mg ml$^{-1}$; Roche) using a GentleMacs dissociator (Miltenyi Biotec) according to manufacturer's instructions. Lungs were incubated at 37 °C for 30 min and further homogenized with the GentleMacs dissociator. The homogenates were filtered through a 70 μm cell strainer (MACS SmartStainers, Miltenyi Biotec). Erythrocytes were lysed with RBC Lysing Buffer for 1 min and single cells were resuspended in PBS with 2% FBS and 2 mM EDTA and stained for surface and intracellular markers. Spleens and lymph nodes were mashed with the back of a syringe in RPMI with 2 mg ml$^{-1}$ Collagenase D and 40 U ml$^{-1}$ DNase I, incubated for 20 min at 37 °C and strained through a 70 μM cell strainer before lysing erythrocytes with RBC Lysing Buffer for 1 min.

## Flow cytometry

Single cells were incubated with mouse Fc receptor blocking reagent (Miltenyi) for 20 min at 4 °C, washed and stained with fluorochrome-conjugated antibodies (at a 1:200 dilution, Supplementary Table 1) for 20 min at 4 °C. Cells were fixed in 4% PFA and analyzed in a Gallios flow cytometer (Beckman Coulter). For intracellular staining, cells were first permeabilized and fixed with the FoxP3 staining set (Miltenyi) according to manufacturer instructions.

For gp33-specific CD8$^+$ T cell staining, after the Fc receptor blocking step, cells were stained with APC-conjugated H-2Db-gp$_{33-41}$ (KAVYNFATC) Dextramer (Immudex) for 10 min at room temperature, and then extracellular antibodies were added and further incubated for 20 min at 4 °C. OVA-specific CD8$^+$ T cells were detected with BV421-conjugated SIINFEKL Tetramer (MBL) following an equivalent protocol.

For spectral flow cytometry, single cells were stained for viability with LIVE/DEAD Fixable Blue for 10 min at room temperature. Cells were then washed with FACS buffer and stained with fluorochrome conjugated antibodies (Supplementary Table 2) diluted in Brilliant staining buffer (BD), in the presence of anti CD16/CD32 Fc Block, for 15 min at 4 °C. Afterwards, cells were washed with FACS buffer, and acquired in an Aurora Cytek 5 L spectral flow cytometer. Uniform Manifold Approximation and Projection (UMAP) analysis was carried out using all markers listed in Supplementary Table 2. Data were manually gated to remove debris, aggregates, LIVE/DEAD-positive dead cells, selecting CD45 positive events, and then downsampled to include equivalent numbers of CD45$^+$ live singlets from each sample, considering the sample with fewer cells as limiting. Subsequently, the UMAP analysis was performed to visualize the different subpopulations in groups. All fluorescent parameters were used besides live and CD45$^+$ cells. UMAP was run using 15 nearest neighbors, a minimal distance of 0.5, in 2-dimensions and Euclidean distance and spectral initialization mode based on the UMAP plugin included in the FlowJo software.

## Re-stimulation and intracellular cytokine staining

For ex vivo stimulation of T cells, lung single cell suspensions were stimulated with 2 μg ml$^{-1}$ of plate bound αCD3 (Miltenyi, clone 145-2C11) and 5 μg ml$^{-1}$ of soluble αCD28 (BD, clone 37.51) for 4 h at 37 °C in the presence of Brefeldin A (eBioscience) in complete RPMI medium containing 50 μM β-mercaptoethanol (Gibco). For ex vivo NK cell stimulation, single cell suspensions were stimulated with 20 μg ml$^{-1}$ of plate bound αNK1.1 (clone PK136) in the presence of FITC-conjugated anti-CD107a antibody (BD, clone 1D4B) and Brefeldin A for 4 h at 37 °C. For DC stimulation, single cell suspensions were stimulated in the presence of 10 ng ml$^{-1}$ of LPS (from E.Coli O111:B4, Sigma Aldrich) and 10 ng ml$^{-1}$ of recombinant mouse IFN-γ (Miltenyi). After stimulation, cell surface staining was carried out as described and intracellular staining was performed with FoxP3 staining set (Miltenyi).

## In vivo CD45 labeling

For intravenous CD45 labeling, mice were intravenously inoculated with 2 μg of CD45-PerCP-Vio700 (Clone REA737, Miltenyi) 5 min before euthanasia and lungs were processed as described.

## In vitro NK cytotoxicity assays

**Mouse studies.** NK cells were isolated from the spleens of mice with anti-CD49b (DX5) microbeads (Miltenyi). Alternatively, NK cells were isolated from the lungs of mice by negative magnetic selection with the EasySep™ Mouse NK Cell Isolation Kit (StemCell). Lung NK cells were expanded in complete RPMI supplemented with 1% non-essential aminoacids (Gibco), 1 mM MEM Sodium Pyruvate (Gibco), 50 μM β-mercaptoethanol and 1000 U ml$^{-1}$ mouse IL-2 (Miltenyi) for 5–7 days. Isolated NK cells were seeded over CellTrace Violet (Invitrogen)-labeled target tumor cells in a 96-well plate at different effector-to-target ratios. After 20 h, dead cells were stained with 7-AAD (Miltenyi) and analyzed by flow cytometry.

**Human studies.** 10$^4$ lung cancer cells (H1322 and H2188) cells were plated in 96-well flat-bottom plates in triplicates. The next day, cells were labeled with calcein-AM (Molecular Probes, C3100MP) and pretreated with HP1F7 antibody to block MHC-I. PBMCs were co-cultured with target cells for 3 h at an E:T ratio of 5:1 (the percentage of NK cells was previously determined for each donor by flow cytometry). Calcein-AM release was determined by measuring absorbance using BioNova® F5 System. Specific lysis was expressed as a percentage, calculated as the ratio [(value − spontaneous release)/(maximum − spontaneous release)] × 100. Spontaneous release corresponds to target cells alone. Maximum release was determined by lysing the target cells in 0.5% Triton X-100 (ThermoScientific). NK cells were identified with the following antibodies: CD3-PacificBlue, CD16-PE-Cy7, and CD56-PE, from Biolegend. For degranulation experiments, cells were additionally stained with LAMP1-APC (Biolegend) for 30 min. After staining, cells were washed in PBS and analyzed in a CytoFLEX (Beckman Coulter). Analysis was performed using Kaluza software and representative gating strategies for CD107a expression on NK cells are shown in Supplementary Fig. 14.

## Splenocyte-mediated cytotoxicity assay

B16-F10-ZsGreenLuc or LLC-ZsGreenLuc cells were seeded in dark 96 well-plates as target cells. Splenocytes from tumor-bearing animals were isolated as described and seeded at a 100:1 ratio over the target cells for 20 h. Then, 150 μg ml$^{-1}$ of Xenolight D-luciferin Potassium Salt (Perkin Elmer) was added to the wells, incubated for 15 min at 37 °C and luminescence was measured in an Epoch Microplate reader (BioTek). % cytotoxicity was calculated with reference to the luminescence emitted by control wells incubated without splenocytes.

## Protein extraction from tissue and serum analysis

Lung tissue was weighted and homogenized in a GentleMacs Dissociator in serum-free RPMI medium using a protein extraction protocol. Then, homogenates were spun at 1500 rpm for 5 min and the supernatant was collected. Then, complete Protease Inhibitor Cocktail (Roche) was added to the supernatants following manufacturer instructions and samples were frozen at −80 °C. CCL5 concentration in

lung lysates was measured by ELISA (Mouse CCL5/RANTES DuoSet, R&D Systems) and was expressed as amount of CCL5 per mg of tissue homogenized.

IL-6 and TNF cytokine concentration was measured in the serum of mice with the ELISA Flex: Mouse IL-6 and ELISA Flex: Mouse TNF kits, following manufacturer instructions (Mabtech). ALP and AST activity and albumin concentration in the serum were measured with a commercial kit (Mammalian Liver Profile #500-0040, Zoetis) in a VetScan VS2 biochemical analyzer (Zoetis).

### Histology and IHC

Formalin fixed paraffin-embedded lung tissue sections (4 µm) mounted in slides were deparaffinized and antigns retrieved by incubation at 95 °C for 30 min in a Sodium Phosphate buffer solution pH 6 (for CD8) or 10 min in a Tris-EDTA pH 8 solution (for NCR1). Endogenous peroxidase was blocked by incubation in 3% $H_2O_2$ solution for 5 minutes, and then sections were further blocked with 3% bovine serum albumin (CD8) or 10% goat serum (NCR1). As primary antibodies, anti-CD8α (clone D4WH2Z, Cell Signal) or anti-NCR1 (ab233558, Abcam) were used at a 1:250 dilution, for 1 h. Primary antibodies were detected with HRP-conjugated anti-rabbit (Envision 4003, Dako) at a 1:20.000 dilution. Chromogenic revelation was carried out with DAB (Dako) and then slides were counterstained with haematoxylin. All the immunohistochemical procedures were performed using an automated autostainer (Autostainer Plus, Dako), and samples were digitalized with a Zeiss AxioScan slide scanner and analyzed using ImageJ software.

### Clonogenic assay

Single-cell suspensions from B16-F10 tumor-bearing lungs were diluted 1:10 or 1:100 in 10 mL of complete DMEM and two replicates were plated for each of the samples in 6-well plates. After 2 weeks, colonies resulting from tumor cell proliferation were fixed with 4% PFA, stained with a 1% (w/v) solution of crystal violet and visually counted.

### Data Analysis

Tumor-bearing mice were randomized into different treatment groups. To avoid cage-associated variability, each cage contained mice from every experimental groups. The number of biological replicates and repetitions for each experiment are indicated in figure legends. Animal monitoring and data analysis were not blinded. Flow cytometry data was analyzed using Weasel software (version 3.0.2) following gating strategies shown in Supplementary Figures 15 and 16. For spectral flow cytometry, after correcting spillover at the Aurora cytometer with the SpectroFlo software, analysis of cellular populations was performed with FlowJo (TreeStar) v10.7.2 software, following gating strategies shown in Supplementary Figure 17. GraphPad Prism software (version 8) was used for graphical representation and statistical analysis. Outliers were not excluded for statistical analysis. All tests applied were two-sided. The group means for different treatments were compared by ANOVA with Bonferroni's multiple comparisons test, or by a two-sided Student's $t$ test. Survival was analyzed by the Log-rank (Mantel-Cox) test. $P$ values < 0.05 were considered significant.

### Reporting summary

Further information on research design is available in the Nature Portfolio Reporting Summary linked to this article.

## Data availability

Raw data is provided as an Excel file in the Source Data section. Materials are available upon request to the authors. Source data are provided with this paper.

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

## Acknowledgements

This work was supported by MCIN/AEI/10.13039/501100011033 [grants number RTI2018-097625-B-I00 and PID2022-138624OB-I00], Gobierno de Aragón [grant number LMP50_21] and Asociación Española Contra el Cáncer (AECC) [grant number IDEAS211042AGUI]. NA was the principal investigator of all these grants. This research was supported by CIBER -Consorcio Centro de Investigación Biomédica en Red- (Groups CB06/06/0020 and CB21/13/00087), Instituto de Salud Carlos III, Ministerio de Ciencia e Innovación and Unión Europea – European Regional Development Fund. DS laboratory is funded by the CNIC; by the European Union's Horizon 2020 research and innovation program under grant agreement ERC-2016-Consolidator Grant 725091; by MCIN PID2019-108157RB MCIN/AEI/10.13039/501100011033 and CPP2021-008310 MCIN/AEI/10.13039/501100011033 Unión Europea Next GenerationEU/PRTR; by Comunidad de Madrid (P2022/BMD-7333 INMU-NOVAR-CM); and by "la Caixa" Foundation (LCF/PR/HR20/00075 and LCF/PR/HR22/00253).The laboratory of C.d.F. is funded by Instituto de Salud Carlos III (ISCIII) through the projects CP20/00106 and PI21/01178 and co-funded by the European Union. The laboratory of M.V-G. is funded by Spanish Ministry of Science and Innovation [grant number PID2021-123795OB-I00]. C.G. has a pre-doctoral fellowship from Gobierno de Aragón. M.J.F. is PhD Candidate at the Autonomous Madrid University. The funders had no role in study design, data collection and analysis, decision to publish or preparation of the manuscript. Authors acknowledge the Scientific and Technical Services from Instituto Aragonés de Ciencias de la Salud (IACS) and Universidad de Zaragoza.

## Author contributions

E.M., M.V-G., C.M., C.d.F., D.S., and N.A. designed the experiments. E.M., A.J-C., I.R-V., C.G., S.U., C.G., A.B.G., P.M-M., L.M., M.A-V., M.J.F., G.E. and I.U-M. performed the experiments. M.A. and J.P. shared animal models. E.M., C.M., M.V-G., C.d.F., D.S., and N.A. wrote the manuscript. N.A. supervised the study.

## Competing interests

The authors declare no competing interests.
