## [Peer Review File · Nature Communications]

REVIEWER COMMENTS

Reviewer #1 (expert in BCG in autoimmune disease):

Comments to the Authors

The title needs to be changed to indicate “in a mouse model”

Also in the abstract, need to be more transparent earlier that this is mouse data

Also in the Discussion, bold statements but you keep forgetting to say “in a mouse model”

The efficacy of BCG to prevent lung cancer has already been described in humans and this important literature is not referenced nor discussed.

JAMA 2019, Sept 4, 2 e1912014, doi: 10.1001/jamanetworkopen.2019.12014

This data is of interest but should discuss the weakness of the data. You administered only 7 days after tumor cell administration, the BCG vaccine. At this stage the tumor could only be observed by histology which means the tumors were very early. Generally, in mouse models, the tumor is grossly visible prior to the therapy intervention to be meaningful. Did you do experiments with mice having more typical gross tumor not just microscopic tumor? If this data is not available need to change title, abstract discussion to indicate in “very early cancer state of microscopic disease....” Could see a benefit of BCG.

Like the data of Naomi Aronson, where BCG prevented lung cancer, did you also try the opposite ie BCG vaccination in mice followed by tumor challenge weeks to months later as a model of prevention of cancer as well? This might be worthwhile to do as well.

Reviewer #2 (expert in tumour Immunology and immune checkpoint therapy):

Comments to the authors

Moreo et al. report the anti-tumor activity of a single intravenous (i.v.) injection of BCG, a live attenuated strain of *Mycobacterium bovis*, in the syngeneic B16-F10 metastatic model. The antitumor effect was further improved by the combination of systemic PD-L1 blockade as compared to i.v. BCG, or PD-L1 blockade alone. Additionally, the authors confirmed the enhanced activity of this combination therapy in the syngeneic Lewis Lung Carcinoma orthotopic model of lung cancer. Functionally, they

found that live BCG -and not heat-killed BCG- activates NK cells cytotoxicity in the lung, which in turn increases the activation of cDC1 in a CCL5-dependent manner, leading to a tumor-specific CD8+ adaptive immune response.

Importantly, the authors confirmed the enhanced anti-tumor efficacy of intravenous BCG as compared to other routes of BCG delivery including intra-nasal, and intradermal injection. In addition, they defined the dose of 1E6 BCG as the minimum dose required to trigger this anti-tumor effect.

However, three critical points need to be addressed in a revised version of the manuscript.

First, the toxicity profile of this therapeutic strategy is not been reported and is dramatically missing. Indeed, potential safety issues associated with i.v. immunization of a live attenuated bacteria, may limit the development of this therapeutic strategy in the clinic.

Therefore, a deep evaluation of the toxicity profile of the monotherapy and combination therapy in each pre-clinical model should be reported. The authors should include the monitoring of body weight overtime following intravenous BCG, together with the absence of bacterial colonization in the lung (normal tissue) and other organs (liver, kidney, distant abdominal lymph nodes, heart).

Those data are critical to fully appreciate the toxicity profile of this therapeutic strategy, and mandatory to further support its evaluation in early phase clinical trials.

Second, a comprehensive immune profiling of the anti-tumor immune response in the lung is lacking (Figure 2). The authors orientate from the onset their analysis/experiments to describe a cellular-mediated adaptive immune response (mostly CD8+). Flow cytometry analysis should include an unbiased evaluation of both myeloid and lymphoid compartments (i.e. unconventional T cells, gamma-delta ($\gamma\delta$) T cells, NK cells, B cells). Of note, recent data showed that intravenous BCG vaccination induced a robust antigen-specific humoral immune response, which correlated with prevention of *Mycobacterium tuberculosis* infection in rhesus macaques (Irvine et al. *Nat. Immunol.* 2021). Furthermore, $\gamma\delta$ T cells are present in the lung mucosa and may also have a role in the antitumor activity of MHC-I deficient malignant cells. Eventually, the cytotoxic activity of CD4+ T cells should be thoroughly investigated, as CD4+ may also have a cytotoxic activity (Oh et al *Cell* 2020).

Third, the distribution pattern of in situ immune infiltration in the lung has not been investigated by the authors. Lung tissues have been dissociated without differentiating lung tumor versus normal adjacent tissue. This segregation should be performed at least in the LLC orthotopic model. Thus, additional in situ staining with CD8/CD4/NK cells and epithelial markers, together with MHC-I staining on malignant cells should be included. Quantifying the immune infiltration in the tumor and the adjacent non tumoral lung tissue should be reported to improve our understanding of the in situ immune response.

Additionally, mycobacteria staining in the lung should be assessed carefully and correlated with the pattern of immune infiltration.

Additional comments

- Discussion

As BCG has been widely used as a vaccine against TB, antibodies against BCG are detected in the serum of the patients (Goubet et al. Cancer Discov 2022). Thus, the efficacy of intravenous BCG in humans may be hampered by the presence of circulating specific immunoglobulins anti-BCG. Please discuss this point.

- Figure 1.

Line 89. At day 7 after tumor cell injection, tumors were already established in the lungs by histological analysis (Fig 1b). Please provide supplementary data, the count of tumor foci at day 7 and day 14. Please also indicate the proportion of mice with tumor graft. Could you provide an additional method to estimate the lung tumor burden, by quantifying the surface area of the tumors in tissue sections.

- Figure 2.

The frequency of IFN γ + CD4+ increases by 5 upon i.v. BCG (Fold change=2 for CD8+). A deep analysis CD4+ cytotoxicity should be included. IFN γ expression should also be evaluated in other cells including NK cells, gamma-delta T cells, and B cells.

- Figure 4.

In situ evaluation of the MHC status on tumor cells should be implemented together with IF staining for NK cells markers. Extended data Fig.1 show the MHC-I status of B16-F10 and LLC untreated cell lines, and their sensitivity to IFN γ (histograms on the left). Interestingly, B16-F10 cells are MHC-I deficient (no difference between FMO and control cells), but sensitive to IFN γ exposure. LLC cancer cells do express MHC-I, which is slightly increased upon IFN γ exposure. The differences in MHC-I expression at baseline may explain the differences in response of PD-L1 blockade monotherapy, shown in Figure 6e and Figure 6f (PBS-black line- and aPD-L1-green line). In the MHC-I proficient LLC tumor model, PD-L1 blockade significantly improves survival as compared to PBS control (Fig.6f). Conversely, aPD-L1 alone did not confer any survival advantage to MHC-I deficient B16F-10 lung tumor model (Fig 6e). The MHC-I status of tumor cells should be evaluated in situ, in addition to NK cells immune infiltration, to provide a meaningful analysis of the interaction/co-localization of both cells in situ. Figure 4f. Provide the gating strategy and dot plots for Lamp1+ expression by NK cells.

- Figure 5. Importantly, the authors show that BCG anti-tumor efficacy relies on a coordination between NK and T cells, through the activation by cDC1 enhanced by CCL5. IF staining showing co-localization of NK and CCL5 markers could be added to validate these data in situ.
- Figure 6. Please include the gating strategy of your myeloid/lymphoid panel in the supplementary data. Did you exclude the Tregs from your analysis of conventional CD4+ T cells. MFI and frequency of PD1 expression on conventional CD4+ and CD8+ T cells should be included in the main figure. PD-L1 expression on tumor cells in vivo/in vitro is lacking and should be assessed.
- Is there a rationale to evaluate the efficacy of PD-L1 blockade rather than PD1 blockade in those models?

Reviewer #3 (expert in thoracic Oncology and NSCLC):

Summary

Interesting, thought-provoking study describing the potential for systemic BCG administration to lead to anti-tumor immunity including possible synergy with immune checkpoint inhibition. Authors do a nice job of sequentially elucidating potential pathways by which systemic BCG administration can lead to anti-tumor immunity, inclusive of NK, T-cell, and dendritic cell roles. All that said, inconsistencies in the type of tumor being evaluated (melanoma vs NSCLC) throughout the manuscript, compounded by the minimal number of models used, significantly dampen the potential impact of the manuscript and leave uncertainty regarding implications of results. As such, the experiments described in the manuscript, remain more of the thought provoking, exploratory nature, rather than full fledged, well-validated results ready for top-tier journal publication.

Major Concern (as above)

-The manuscript suggests stimulation of anti-tumor lung immunity, which seems misleading, since the cell line studied is a melanoma cell line that metastasizes to the lungs. This is somewhat mitigated by the use of lung cancer lines H2228 and H3122, as well as Lewis Lung Carcinoma cells, but switching b/w melanoma and lung cancer without clear rationale creates concerns re: consistency of findings. Authors need to explain why such an approach was undertaken.

Minor Comments

-Phrasing is choppy at some points in the manuscript, with missed words, hard to understand descriptions, and run-on sentences. This is a minor concern, but certainly would require significant editorial reformatting prior to publication.

Answers to Prompts

What are the noteworthy results?

Systemic BCG administration could improve anti-tumor immunity in solid tumors (melanoma and/or NSCLC).

Will the work be of significance to the field and related fields?

As above, this manuscript seems more exploratory and, as such, is not complete enough to have significant impact on the field of lung cancer research at this time.

How does it compare to the established literature? If the work is not original, please provide relevant references.

It is novel, and authors do a nice job of contextualizing their findings, but given how novel it is in the lung cancer field, requires a more rigorous evaluation to impact the field and convince this is a highly promising pathway that requires further attention.

Does the work support the conclusions and claims, or is additional evidence needed?

As above, using both melanoma and lung cancer models is not a good approach, when the goal is to evaluate lung cancer. Additional, consistent evidence in lung cancer specific models is required for this manuscript to be elevated to level of high impact.

Are there any flaws in the data analysis, interpretation and conclusions? Do these prohibit publication or require revision?

As above, if authors are studying lung cancer they need to use lung cancer models, not melanoma models, unless explicit, convincing rationale is provided.

Is the methodology sound? Does the work meet the expected standards in your field?

It is sound, but not complete enough to warrant a high-impact on the field in its current form.

Is there enough detail provided in the methods for the work to be reproduced?

Yes, for the experiments described sufficient detail is present

POINT-BY-POINT REPLY LETTER.
MANUSCRIPT NUMBER NCOMMS-22-39644-T.

We appreciate the helpful comments and constructive criticisms of the reviewers. We trust that we have been able to respond to their suggestions and concerns meeting their expectations. In continuation, **a list of the new experiments performed in the revised manuscript:**

1. Assessment of IV BCG efficacy administered at 14 days post-tumor cell challenge, to evaluate efficacy of BCG in a high tumor burden setting. *Figures 1 and 5*
2. Histological analyses of lung tumors. HE images from tumor sections were obtained at different timepoints and number of tumor nodules and their area were determined as an additional readout of BCG efficacy, as well as to confirm the presence of tumors at the time of BCG administration. These analyses were done in all the tumor models evaluated in the study. *Figures 1 and 5, and extended figures 1, 2, 9, 10 and 13.*
3. Prophylactic efficacy of IV BCG. In this case, mice were vaccinated with intravenous or subcutaneous BCG 42 days prior to tumor challenge to evaluate vaccine capacity to prevent tumor implementation. The experiment was repeated in *IFN γ ^{-/-}* mice. *Extended data figures 4 and 10.*
4. Tolerability and biodistribution of IV BCG. In this experiment mice were treated with IV BCG and different safety parameters were monitored to detect toxicity. In addition, serum samples were taken to analyze liver function and inflammatory cytokines. The experiments were done in tumor-free and tumor-bearing mice. In the former, we added a group treated with both BCG and anti-PD-L1. *Extended data figures 3, 11 and 12.*
5. Biodistribution of IV BCG. Mice were treated with IV BCG and bacterial presence was determined in different lymphoid and non-lymphoid organs at 2 months post-immunization. *Extended data figure 3.*
6. Global analysis of immune populations in tumor-bearing lungs. Mice were challenged with B16-F10 cells at day 20, treated or not with BCG. Multiparameter analysis of immunological populations was conducted by spectral flow cytometry. *Extended data figure 5.* Some of these populations were also analyzed on LLC experiments by conventional flow cytometry. *Figure 6.*
7. Assessment of the *in vivo* functionality of the tumor-specific immune response induced by BCG. Splenocytes from BCG-treated tumor-bearing lungs were harvested and transferred to *Rag1^{-/-}* mice, which were subsequently challenged subcutaneously with the tumor cells to determine the capacity of transferred cells to impair tumor growth in comparison to mice

transferred with splenocytes from untreated tumor-bearing mice, or from BCG-treated tumor-naïve mice. *Figure 2.*

8. *In situ* characterization of CD8⁺T cells and NK cells contained in the LLC tumors. Lung tissue sections were stained by IHC and tumor-infiltrated CD8⁺ T cells and NK cells quantified. *Figure 6.*
9. Influence of pre-existing BCG-specific immunity on IV BCG antitumoral efficacy. Mice were vaccinated by subcutaneous route and 42 days later challenged with B16-F10 cells, and treated subsequently at day 7 post-tumor implementation with IV BCG. *Extended data figure 4.*
10. Evaluation IV BCG efficacy in an additional lung cancer orthotopic model induced by IV administration of TC-1 cells. *Figure 5 and extended data figure 10.*
11. Analysis of tumor-specific CD8⁺T cell responses in the LLC model. This experiment was performed using an OVA-expressing LLC cell line. *Figure 6 and extended data figure 10.*
12. Evaluation of cytotoxic immune response in mice treated with combination of IV BCG and anti-PD-L1. *Figure 7.*

In accordance to the previous and new experiments, **the main conclusions of the study are:**

1. IV BCG induces prophylactic and therapeutic antitumoral efficacy in different murine models of lung melanoma metastasis and lung cancer.
2. A single dose of IV BCG (10⁶ CFUs) is well tolerated and does not trigger evident toxicity, at least in any of the parameters tested.
3. IV BCG induces functional tumor-specific CD8⁺ T cell responses, and activates NK cells *in vivo*. In addition, IV BCG increases the infiltration of both NK cells and CD8⁺ T lymphocytes into the tumor.
4. Depletion of NK cells, cDC1s or T cells abrogates IV BCG efficacy. Mechanistically, IV BCG activates NK cells, which enhance cDC1 migration to lungs and kill tumor cells providing tumor antigens to cDC1 for subsequent antigen presentation and generation of tumor-specific adaptive immune responses.
5. IV BCG synergizes with anti-PD-L1 treatment in both B16-F10 and LLC tumor models. Remarkably, in both models anti-PD-L1 used as monotherapy has a modest therapeutic effect, suggesting that IV BCG can sensitize tumors to immune checkpoint blockade.

Next, we will respond **point-by-point** to the different questions and concerns raised by the reviewers (comments in bold and *italics*).

REVIEWER 1

1. The title needs to be changed to indicate “in a mouse model”. Also in the abstract, need to be more transparent earlier that this is mouse data. Also in the Discussion, bold statements but you keep forgetting to say “in a mouse model”.

RESPONSE: Title has been amended in the new version, incorporating the information that experiments were done in mice. With regard to abstract and discussion, we have also specified in several occasions that the study was conducted in murine cancer models. We consider that the revised manuscript is not confusing on this aspect, and a potential future reader will understand that the conclusions obtained were generated with experiments conducted in mice.

2. The efficacy of BCG to prevent lung cancer has already been described in humans and this important literature is not referenced nor discussed. JAMA 2019, Sept 4, 2 e1912014, doi: 10.1001/jamanetworkopen.2019.12014

RESPONSE: We agree with the reviewer in the importance of this study in the context of our results, as they demonstrate the prophylactic effect of BCG (intradermal) in humans against lung cancer. The study is now cited in the manuscript.

3. This data is of interest but should discuss the weakness of the data. You administered only 7 days after tumor cell administration, the BCG vaccine. At this stage the tumor could only be observed my histology which means the tumors were very early. Generally, in mouse models, the tumor is grossly visible prior to the therapy intervention to be meaningful. Did you do experiments with mice having more typical gross tumor not just microscopic tumor? If this data is not available need to change title, abstract discussion to indicate in “very early cancer state of microscopic disease....” Could see a benefit of BCG.

RESPONSE: We agree with the reviewer in the importance of the aspect addressed on this comment, as the efficacy of immunotherapy can be compromised in higher tumor burden scenarios. Since we aim to characterize IV BCG as a therapeutic approach we need to ensure that the vaccine is administered over established tumors. Thus, we conducted a new experiment in which we delayed treatment until day 14, a timepoint in which metastatic nodules were grossly visible (Extended Fig. 1 and 9). This experimental setting was used with B16-F10 (Fig. 1c) and LLC (Fig. 5d) tumor models, finding in both cases that the therapeutic benefit conferred by IV BCG was maintained under these more stringent conditions. The result is summarized in the next figure 1.

Figure 1. Therapeutic effect of IV BCG vaccination at 14 days post tumor implantation in B16-F10 (left) and LLC (right) tumor models.

In addition, we have made an important effort to characterize the tumor burden in the lungs at the time of BCG administration and at subsequent timepoints. In the case of B16-F10 cells, whose tumor foci can be easily detected due to melanin expression nodules can be already observed macroscopically at 7 days, and histological analysis confirmed the presence of tumor nodules at this timepoint in all the mice challenged (Extended data fig.1). This can be extrapolated to the other lung cancer models used: LLC (extended data fig.9) and TC-1 (extended data fig. 10). We consider that presence of well-defined nodules in the lungs is an indicator of established tumors.

4. Like the data of Naomi Aronson, where BCG prevented lung cancer, did you also try the opposite ie BCG vaccination in mice followed by tumor challenge weeks to months later as a model of prevention of cancer as well? This might be worthwhile to do as well.

RESPONSE: As suggested by the reviewer, we conducted a new experiment in which we administered BCG (comparing subcutaneous and intravenous routes) 42 days before tumor cell challenge, to evaluate the efficacy of BCG in a prophylactic scenario. Our results with SC BCG were in concordance with the data published in the paper of Naomi Aronson, and showed an increase of mouse survival, from a median of 21 days to 28. In the case of prophylactic IV BCG, vaccination completely prevented B16-F10 tumor growth in 5 out of 6 mice. Additionally, we found that this effect completely relied on host IFN- γ expression, since protection was no longer observed in mice lacking this cytokine. The results were confirmed with the LLC cells. These data are displayed in the revised manuscript in the new extended fig. 4a for B16-F10 cells and extended fig. 10a for LLC. The result is also shown in the next figure 2. Although further characterization of the prophylactic effect of SC and IV BCG in these models is of great interest and the mechanism behind merits further investigation, we think that it falls out of the scope of this work, which is focused on the therapeutic administration of BCG over established tumors.

Figure 2. Prophylactic effect of subcutaneous and intravenous BCG vaccination. Mice were vaccinated with BCG 42 days prior to B16-F10 intravenous challenge.

REVIEWER 2

Moreo et al. report the anti-tumor activity of a single intravenous (i.v.) injection of BCG, a live attenuated strain of Mycobacterium bovis, in the syngeneic B16-F10 metastatic model. The antitumor effect was further improved by the combination of systemic PD-L1 blockade as compared to i.v. BCG, or PD-L1 blockade alone. Additionally, the authors confirmed the enhanced activity of this combination therapy in the syngeneic Lewis Lung Carcinoma orthotopic model of lung cancer. Functionally, they found that live BCG -and not heat-killed BCG- activates NK cells cytotoxicity in the lung, which in turn increases the activation of cDC1 in a CCL5-dependent manner, leading to a tumor-specific CD8+ adaptive immune response.

Importantly, the authors confirmed the enhanced anti-tumor efficacy of intravenous BCG as compared to other routes of BCG delivery including intra-nasal, and intradermal injection. In addition, they defined the dose of 1E6 BCG as the minimum dose required to trigger this anti-tumor effect.

RESPONSE: We thank the reviewer for his/her precise summary of the study.

However, three critical points need to be addressed in a revised version of the manuscript.

1. First, the toxicity profile of this therapeutic strategy is not been reported and is dramatically missing. Indeed, potential safety issues associated with i.v. immunization of a live attenuated bacteria, may limit the development of this therapeutic strategy in the clinic.

Therefore, a deep evaluation of the toxicity profile of the monotherapy and combination therapy in each pre-clinical model should be reported. The authors should include the monitoring of body weight overtime following intravenous BCG, together with the absence of bacterial colonization in the lung (normal tissue) and other organs (liver, kidney, distant abdominal lymph nodes, heart).

Those data are critical to fully appreciate the toxicity profile of this therapeutic strategy, and mandatory to further support its evaluation in early phase clinical trials.

RESPONSE: We agree with the reviewer that safety and toxicity are critical points in the context of our study, considering that we are proposing the administration of live bacteria by intravenous route. In this regard, we performed different experiments to specifically address the tolerability of IV BCG, both in tumor-free and LLC tumor-bearing mice. In the case of tumor-free mice, we also included an additional group in which we assessed IV BCG toxicity in combination with anti-PD-L1. Our results, described in extended data fig. 3, 11 and 12, revealed no sign that could denote the induction of acute toxicity by BCG, including changes of temperature or loss of body weight. We also analyzed some biochemical parameters in serum to

assess liver function, as ALT or ALP activity and albumin concentration, finding no differences with untreated controls. Finally, we also measured inflammatory cytokines IL-6 and TNF α in serum, with no big differences with untreated controls, suggesting that BCG is not triggering sepsis-like systemic inflammatory reactions.

Additionally, as proposed by the reviewer, we studied vaccine biodistribution in different lymphoid and non-lymphoid organs at two months post-immunization. BCG was detected in the spleen, liver, lung and mediastinal and abdominal lymph nodes. but not in other organs as heart or brain. These results are consistent with previous studies¹, in which IV BCG has been reported to distribute to different organs where the bacteria have a limited persistence and are progressively cleared by the immune system. Indeed, our results suggest that this temporal persistence might be important for protection, considering the poorer efficacy shown with lower BCG doses or inactivated bacteria.

In the context of this comment, it is also relevant to mention all the information published in the literature regarding the studies conducted with intravenous BCG in different animal models. IV BCG has been tested in mice^{2,3}, dogs⁴, guinea pigs⁵ and non-human primates⁶, and the conclusions drawn from all of these studies indicate that IV BCG was well-tolerated by the animals. Also remarkable is the existence of clinical trials where other intravenous live attenuated bacteria, as *Salmonella* and *Listeria*, have been tested in humans^{7,8}. This argument has been included in the revised discussion.

2. Second, a comprehensive immune profiling of the anti-tumor immune response in the lung is lacking (Figure 2). The authors orientate from the onset their analysis/experiments to describe a cellular-mediated adaptive immune response (mostly CD8+). Flow cytometry analysis should include an unbiased evaluation of both myeloid and lymphoid compartments (i.e. unconventional T cells, gamma-delta (gd) T cells, NK cells, B cells). Of note, recent data showed that intravenous BCG vaccination induced a robust antigen-specific humoral immune response, which correlated with prevention of Mycobacterium tuberculosis infection in rhesus macaques (Irvine et al. Nat. Immunol. 2021). Furthermore, gd T cells are present in the lung mucosa and may also have a role in the antitumor activity of MHC-I deficient malignant cells. Eventually, the cytotoxic activity of CD4+ T cells should be thoroughly investigated, as CD4+ may also have a cytotoxic activity (Oh et al Cell 2020).

RESPONSE: Following reviewer's suggestion, in the revised manuscript we performed a global characterization of the lung immune compartment from B16-F10 tumor-bearing mice, both untreated and treated with BCG. These results are displayed in the new extended data fig. 5, showing the BCG-mediated increment of DCs and NK cells, but also an increase in interstitial

monocytes/macrophages with an enhanced expression of MHC-II, as well as T cells with an effector phenotype.

With regard to the humoral response, we agree that the role of this type of response is getting more relevancy in cancer immunosurveillance and immunotherapy as more studies about this topic are published. However, the scope of our study is focused on other type of responses that seem to be particularly more relevant for the antitumoral response mediated by IV BCG, such as NK cells and cytotoxic T lymphocytes. Therefore, even though we cannot discard a role of humoral response on protection mediated by IV BCG, this is something that we have not addressed on this manuscript.

In the case of gamma-delta T cells, we found an increased expression of granzyme B after IV BCG treatment. This result is displayed on fig. 6c and mentioned in the results section.

Finally, our new results indicate that CD4+ T cells express more granzyme B upon IV BCG treatment (Fig.6c), which together with our previous observation that depletion of CD4+ T lymphocytes abrogates BCG antitumoral effect suggest a possible contribution cytotoxic CD4+ cells, in addition to their likely contribution as “helper” cells. However, we also found in the characterization experiments of B16-F10 and LLC tumor cells, that these cell lines completely lack MHC-II expression, even when exposed to IFN γ , meaning that direct recognition of tumor cells is not likely the mechanism by which CD4+ T cells participate in IV BCG efficacy, at least in this specific models. This lack of MHC-II expression on LLC tumor cells is consistent with the literature⁹.

3. Third, the distribution pattern of in situ immune infiltration in the lung has not been investigated by the authors. Lung tissues have been dissociated without differentiating lung tumor versus normal adjacent tissue. This segregation should be performed at least in the LLC orthotopic model. Thus, additional in situ staining with CD8/CD4/NK cells and epithelial markers, together with MHC-I staining on malignant cells should be included. Quantifying the immune infiltration in the tumor and the adjacent non tumoral lung tissue should be reported to improve our understanding of the in situ immune response. Additionally, mycobacteria staining in the lung should be assessed carefully and correlated with the pattern of immune infiltration.

RESPONSE: We thank the reviewer for this comment, since distribution pattern of cytotoxic cells within the tumors is an aspect not covered in the previous manuscript version. In the revised version, we have analyzed by IHC the presence of NK cells and CD8+ T lymphocytes in tissue sections from LLC tumor-bearing lungs, both inside the tumor and in the adjacent areas. These results are included in the new fig. 6g, h. Of note, we found a substantial increment in both

CD8⁺ and NK cells inside the tumors from BCG-treated mice. These results, together with the functional assays performed and the *in vivo* neutralization experiments conducted, clearly support our conclusions regarding the crucial role of CD8⁺ T and NK cells for the antitumoral efficacy of IV BCG.

Additionally, following reviewer's advice we conducted an experiment treating the mice with IV GFP-expressing BCG, with the aim to detect them by microscopy inside tumors. We previously conducted an experiment using a fluorescent BCG version, where lung suspensions were obtained to analyze infected cells by flow cytometry. Our results indicated that BCG-infected cells could be identified, and bacteria were mainly contained in macrophage populations, both alveolar (CD11c⁺SiglecF⁺) and interstitial (CD11c⁺SiglecF⁻CD64⁺). These results are summarized in the next figure:

Figure 3. Infection of lung macrophages by BCG following IV administration. Mice were treated intravenously with GFP-expressing BCG and GFP-positive cells analyzed in lung cellular suspensions, finding that the whole infected compartment correspond to macrophages.

However, when we tried to identify infected cells (GFP⁺) on fixed lung sections by microscopy, we were unable to obtain any positive results. Our hypothesis (based on our experience) is that the sensitivity of finding bacteria by microscopy in lung tissue sections is much lower than by flow cytometry. The reason is because by microscopy we do not analyze the whole lung as by flow cytometry, and the probability to find the bacteria in a concrete section of tissue is very low, especially when bacterial numbers are not very high. In our experience with *in vivo* infections using virulent tuberculosis bacteria, bacilli can be identified in tissue sections in conditions in which bacteria replicate up to 10⁷ bacilli per animal, but it is much more difficult at lower bacterial burdens. Therefore, with the experimental settings used in the present study, where we do not expect to reach more than 10⁴ CFUs at the timepoint studied, the probability to find BCG is much lower, which it would explain the divergence between flow cytometry and microscopy analyses.

Even though we consider that the topic addressed on this comment is very interesting, we have not included the result obtained by flow cytometry in the manuscript as we consider that the

data are incomplete without the analysis of the *in situ* distribution of infected cells. We plan to work on the optimization of this technique to increase the limit of detection, with the aim to obtain this characterization for a future study in which we are working at present, based on the specific contribution of macrophages to IV BCG therapeutic efficacy.

Additional comments

Discussion

4. As BCG has been widely used as a vaccine against TB, antibodies against BCG are detected in the serum of the patients (Goubet et al. Cancer Discov 2022). Thus, the efficacy of intravenous BCG in humans may be hampered by the presence of circulating specific immunoglobulins anti-BCG. Please discuss this point.

RESPONSE: We find that the point addressed by the reviewer is very relevant. Looking at the literature, there are different studies assessing whether previous BCG immunization blocks or boosts a subsequent immunization with the bacteria. The general conclusion of these studies is that presence of a pre-existing BCG-specific response enhances the immune response induced by a second contact with the bacteria. This has been observed both in the context of bladder cancer treatment¹⁰ and tuberculosis vaccination¹¹.

In the context of our study, we conducted an experiment in which we administered IV BCG over tumor-bearing mice previously immunized or not with BCG by subcutaneous route (which mimics the intradermal route used in clinic). The result is shown in the figure below and in the current extended data fig. 4c from the manuscript, and agrees with previous literature with regard to the beneficial effect of previous BCG vaccination to enhance the efficacy of IV BCG.

Figure 4. Influence of pre-existing BCG immunity on IV BCG antitumoral efficacy. Mice were subcutaneously vaccinated with BCG, and 42 days later challenged with B16-F10 cells and treated with IV BCG.

5. Figure 1. Line 89. At day 7 after tumor cell injection, tumors were already established in the lungs by histological analysis (Fig 1b). Please provide extended data, the count of tumor foci at day 7 and day 14. Please also indicate the proportion of mice with tumor graft. Could you provide an additional method to estimate the lung tumor burden, by quantifying the surface area of the tumors in tissue sections.

RESPONSE: A complete histological analysis of the three tumor models used in the revised manuscript: B16-F10, LLC and TC-1, has been performed, including days 7 and 14, and they are described in the fig. 1 and 5, and the extended data fig. 1, 2, 9, 10 and 13. The count of tumor nodules, proportion of mice with tumor graft and tumor surface areas have been included as alternative methods to determine lung tumor burden. Additionally, we also conducted clonogenic assays of lung single cell suspensions to assess how IV BCG treatment influences the clonogenic capacity of tumor cells. These results are also included in the figures referred above.

6. Figure 2. The frequency of IFN γ + CD4+ increases by 5 upon i.v. BCG (Fold change=2 for CD8+). A deep analysis CD4+ cytotoxicity should be included. IFN γ expression should also be evaluated in other cells including NK cells, gamma-delta T cells, and B cells.

RESPONSE: Our new results indicate that CD4+ T cells express more granzyme B in the BCG-treated animals (fig. 6c), which together with our previous observation that depletion of CD4+ T lymphocytes abrogates BCG antitumoral effect suggest a possible contribution cytotoxic CD4+ cells, in addition to their likely contribution as “helper” cells. We mention this possibility in the revised manuscript.

In addition, since NK cells play a crucial role in the efficacy mechanism of IV BCG, confirmed by functional killing assays and *in vivo* NK cell neutralization experiments, IFN γ production and other activation markers were analyzed specifically over NK cells (fig. 4). With regard to gamma-delta T cells and B cells, their frequencies were determined in the new analysis of the immune populations conducted by spectral flow cytometry (extended data fig. 5), and in the case of gamma-delta T cells we have seen an increase of granzyme B expression on these cells induced by BCG (fig. 6c), which suggests a possible contribution to cytotoxicity against tumor cells. Although we agree that a deeper characterization of B cells and gd lymphocytes would be highly interesting, no other assessments have been done specifically over these cellular subsets, as the study is mainly focused on NK cells and conventional T lymphocytes.

7. Figure 4. In situ evaluation of the MHC status on tumor cells should be implemented together with IF staining for NK cells markers. Extended data Fig.1 show the MHC-I status of B16-F10 and LLC untreated cell lines, and their sensitivity to IFN γ (histograms on the left). Interestingly, B16-F10 cells are MHC-I deficient (no difference between FMO and control cells), but sensitive to IFN γ exposure. LLC cancer cells do express MHC-I, which is slightly increased upon IFN γ exposure. The differences in MHC-I expression at baseline may explain the differences in response of PD-L1 blockade monotherapy, shown in Figure 6e and Figure 6f (PBS-black line- and aPD-L1-green line). In the MHC-I proficient LLC tumor model, PD-L1 blockade significantly improves survival as compared to PBS control (Fig.6f). Conversely,

aPD-L1 alone did not confer any survival advantage to MHC-I deficient B16F-10 lung tumor model (Fig 6e). The MHC-I status of tumor cells should be evaluated in situ, in addition to NK cells immune infiltration, to provide a meaningful analysis of the interaction/co-localization of both cells in situ. Figure 4f. Provide the gating strategy and dot plots for Lamp1+ expression by NK cells.

RESPONSE: As explained above, we agree with the reviewer in the importance of evaluating *in situ* the presence of NK cells within the tumor. This has been specifically assessed in the revised manuscript, finding the IV BCG enhances the infiltration of NK cells into tumor environment (fig. 6g). These results, together with the lack of efficacy against B16-F10 tumors observed following NK cell depletion and in perforin knockout mice, as well as experiments with MHC-I-lacking B16-F10 cells (B16-F10-*B2m*^{-/-}), clearly suggest that BCG-activated NK cells are detecting and killing B16-F10 cells *in vivo*, although ultimately T cells are needed in the case of MHC-I sufficient tumors. Therefore, although we find that the proposal of the reviewer is interesting, we have not conducted IF staining studies to colocalize NK cells and MHC-I-expressing tumor cells, as we consider that this point has been properly addressed with the different experiments conducted, and the conclusions in this regard are robust.

In addition, we find very interesting the observation of the reviewer about the differences of MHC-I basal expression between B16-F10 and LLC, and the differential sensitivity to IFN γ -induced response. Indeed, this is something that we are studying in another project, with the aim to determine how these differences can determine sensitivity to ICIs. In the current manuscript, focused on IV BCG, we have seen that both B16-F10 and LLC are sensitive to IV BCG (both alone or in combination with antiPD-L1), regardless of the different MHC-I basal expression between both cell lines. We hypothesize that BCG-driven IFN γ production likely enhances MHC-I expression on B16-F10 cells, making them more sensitive to CD8⁺ cell-mediated recognition and elimination. Supporting this hypothesis, we challenged mice intravenously with fluorescent ZsGreen-expressing B16-F10 cells, and found that MHC-I expression on lung tumor cells is augmented in mice treated with IV BCG. Interestingly, we found that NK cells are required for this upregulation, suggesting that NK cell-derived IFN γ could be mediating MHC-I upregulation on tumor cells. These results are displayed in the following figure 5, but not in the manuscript, as we consider that they do not crucially contribute to support manuscript conclusions.

Figure 5. *In vivo* Expression of MHC-I on lung tumor cells. Mice were challenged with fluorescent ZsGreen-B16-F10 cells, and MHC-I expression analyzed by flow cytometry in tumor cells from lung suspensions.

Finally, gating strategy and dot plots for Lamp1+ expression from killing assays with human NK cells are provided (extended fig. 15)

8. Figure 5. Importantly, the authors show that BCG anti-tumor efficacy relies on a coordination between NK and T cells, through the activation by cDC1 enhanced by CCL5. IF staining showing co-localization of NK and CCL5 markers could be added to validate these data *in situ*.

RESPONSE: In the revised manuscript, we have performed IHC analysis to determine and quantify the presence of NK cells within the tumors, finding that BCG clearly enhances the migration of these cells to the tumor environment (fig. 6g, h). Our results by ICS staining with a specific antiCCL5 antibody indicate that NK cells are major producers of CCL5 (fig. 4k), and quantification of CCL5 levels in lung homogenates resulted increased following BCG treatment in a NK cell-dependent manner (fig. 4l). Thus, considering the data from literature¹² we hypothesize that CCL5 could be mediating cDC1 recruitment. However, we are aware that on this point we only demonstrate a correlation, and not a direct link, for which *in vivo* CCL5 neutralization studies should be performed. This lack of functional confirmation is now mentioned in the discussion section of the study.

9. Figure 6. Please include the gating strategy of your myeloid/lymphoid panel in the extended data. Did you exclude the Tregs from your analysis of conventional CD4+ T cells. MFI and frequency of PD1 expression on conventional CD4+ and CD8+ T cells should be included in the main figure. PD-L1 expression on tumor cells *in vivo/in vitro* is lacking and should be assessed.

RESPONSE: Gating strategies for all the flow cytometry analyses are now included in the extended data (extended data fig. 15, 16, 17 and 18). Responding to reviewer, Tregs were not specifically excluded from CD4+ T cell analyses. MFI and frequency from PD1 expression on T cells, NK cells and gd cells are included in the global analysis of the immune populations performed and shown in the extended data fig. 5. No significant differences between untreated and treated groups were found when we studied PD1 expression on global T cell populations with

B16-F10 cells. In addition, PD-L1 expression on B16-F10 cells is analyzed *in vivo* and shown in fig. 7d, finding that treatment with IV BCG has a limited effect on tumor cell PD-L1.

10. Is there a rationale to evaluate the efficacy of PD-L1 blockade rather than PD1 blockade in those models?

RESPONSE: Both PD-1 and PD-L1 molecules are relevant from a clinical point of view, and specific therapies against both of them are in clinic nowadays. Therefore, we did not have a particular rationale to choose PD-L1 rather than the use of a relevant target to block this interaction. In addition, the neutralizing activity of the anti-PD-L1 clone used in our study has been widely confirmed in the literature. However, we agree that it will be interesting in the future to confirm our results using anti-PD-1 antibodies.

REVIEWER 3

Summary

Interesting, thought-provoking study describing the potential for systemic BCG administration to lead to anti-tumor immunity including possible synergy with immune checkpoint inhibition. Authors do a nice job of sequentially elucidating potential pathways by which systemic BCG administration can lead to anti-tumor immunity, inclusive of NK, T-cell, and dendritic cell roles. All that said, inconsistencies in the type of tumor being evaluated (melanoma vs NSCLC) throughout the manuscript, compounded by the minimal number of models used, significantly dampen the potential impact of the manuscript and leave uncertainty regarding implications of results. As such, the experiments described in the manuscript, remain more of the thought provoking, exploratory nature, rather than full fledged, well-validated results ready for top-tier journal publication.

RESPONSE: We thank the reviewer for this precise summary. We also understand his/her criticism regarding the tumor models tested, and the unbalanced use of melanoma vs lung cancer cells throughout the study. We address this question in the answer to the next comment.

Major Concern (as above)

1. The manuscript suggests stimulation of anti-tumor lung immunity, which seems misleading, since the cell line studied is a melanoma cell line that metastasizes to the lungs. This is somewhat mitigated by the use of lung cancer lines H2228 and H3122, as well as Lewis Lung Carcinoma cells, but switching b/w melanoma and lung cancer without clear rationale creates concerns re: consistency of findings. Authors need to explain why such an approach was undertaken.

RESPONSE: We thank the reviewer for this comment. First, we would like to explain our rationale to study both metastatic melanoma and lung cancer models. Lung cancer is one of the most prevalent type of cancer, as well as the one with highest mortality rates. In addition, the lung is a common site of metastasis for other primary tumors, as breast, colon, melanoma or kidney. Therefore, we consider that it is well justified to evaluate the therapeutic efficacy of BCG against both metastatic and orthotopic lung tumors. In the current version, we have tried to clarify our approach, especially in the introduction section where we include the motivations of the study.

With respect to the comment addressed by the reviewer, we agree with him/her that in the original version of the manuscript the use of B16-F10 melanoma and LLC lung cancer tumor models was unbalanced, with most of the studies conducted in the melanoma model, while extrapolating the conclusions to lung cancer without strong experimental evidence. In the revised manuscript, we have made an effort by conducting new experiments with orthotopic lung cancer

models to extend our findings to NSCLC. Now, there are three specific main figures (fig. 5 and 6 and half of the 7), and 5 extended figures (extended data fig. 9-13), in which lung cancer models are characterized in the context of IV BCG treatment. Importantly, we have tested IV BCG efficacy in a second model of orthotopic lung cancer induced by intravenous administration of the lung tumor cell line TC-1, which has been previously used as a model of NSCLC¹³. In continuation, a summary of the new data is provided:

- Assessment of IV BCG efficacy in lung cancer models by evaluation of tumor area in tissue sections at day 20 of experiment
- Assessment of IV BCG prophylactic efficacy when administered prior to LLC tumor challenge
- Efficacy of IV BCG against a second lung cancer model.
- Analysis of frequencies of NK cells and T lymphocytes in LLC tumor-bearing lungs, as well as functional assays of these cellular subsets based on IFN γ production and killing assays.
- Use of OVA-expressing LLC cells to analyze tumor-specific responses induced by BCG
- IHC and microscopy analyses to assess *in situ* the presence of CD8+ and NK cells in the tumor environment.
- Safety profile of IV BCG in LLC tumor-bearing mice.

As it can be observed in the revised manuscript, all these new data clearly support the conclusions of the study, making them more robust than in the original version. Importantly, at present the use of the metastatic and orthotopic tumor models is more balanced, with most of the experiments conducted in both models.

Summarizing the new findings, we have seen that immunological pathways involved in BCG antitumoral response against LLC tumors are comparable to the described with B16-F10 tumors. IV BCG recruits and activates NK and CD8+ T cells, and strongly enhances their infiltration into the tumor mass. Furthermore, by using a OVA-expressing LLC tumor cell line, we describe that IV BCG also potentiates tumor-specific responses in this model.

Minor Comments

2. Phrasing is choppy at some points in the manuscript, with missed words, hard to understand descriptions, and. This is a minor concern, but certainly would require significant editorial reformatting prior to publication.

RESPONSE: We have revised the manuscript trying to identify the sentences referred by the reviewer and amend it to make the text more readable.

Answers to Prompts

What are the noteworthy results?

Systemic BCG administration could improve anti-tumor immunity in solid tumors (melanoma and/or NSCLC).

Will the work be of significance to the field and related fields?

As above, this manuscript seems more exploratory and, as such, is not complete enough to have significant impact on the field of lung cancer research at this time.

How does it compare to the established literature? If the work is not original, please provide relevant references.

It is novel, and authors do a nice job of contextualizing their findings, but given how novel it is in the lung cancer field, requires a more rigorous evaluation to impact the field and convince this is a highly promising pathway that requires further attention.

Does the work support the conclusions and claims, or is additional evidence needed?

As above, using both melanoma and lung cancer models is not a good approach, when the goal is to evaluate lung cancer. Additional, consistent evidence in lung cancer specific models is required for this manuscript to be elevated to level of high impact.

Are there any flaws in the data analysis, interpretation and conclusions? Do these prohibit publication or require revision?

As above, if authors are studying lung cancer they need to use lung cancer models, not melanoma models, unless explicit, convincing rationale is provided.

Is the methodology sound? Does the work meet the expected standards in your field?

It is sound, but not complete enough to warrant a high-impact on the field in its current form.

Is there enough detail provided in the methods for the work to be reproduced?

Yes, for the experiments described sufficient detail is present

RESPONSE: We thank the reviewer for his/her considerations about our study. We expect to have met his/her expectations with this revised manuscript

REFERENCES

1. Mittrücker, H.-W. *et al.* Poor correlation between BCG vaccination-induced T cell responses and protection against tuberculosis. *Proceedings of the National Academy of Sciences* **104**, 12434–12439 (2007).
2. Kaufmann, E. *et al.* BCG Educates Hematopoietic Stem Cells to Generate Protective Innate Immunity against Tuberculosis. *Cell* **172**, 176–190.e19 (2018).
3. Hilligan, K. L. *et al.* Intravenous administration of BCG protects mice against lethal SARS-CoV-2 challenge. *Journal of Experimental Medicine* **219**, (2022).
4. Owen, L. N. & Bostock, D. E. Effects of Intravenous BCG in Normal Dogs and in Dogs with Spontaneous Osteosarcoma*. *J. Cancer* vol. 10 775–780 Preprint at (1974).
5. Muggleton, P. W., Prince, GillianH. & Hilton, MarjorieL. EFFECT OF INTRAVENOUS B.C.G. IN GUINEAPIGS AND PERTINENCE TO CANCER IMMUNOTHERAPY IN MAN. *The Lancet* **305**, 1353–1355 (1975).
6. Darrah, P. A. *et al.* Prevention of tuberculosis in macaques after intravenous BCG immunization. *Nature* **577**, 95–102 (2020).
7. Brahmer, J. R. *et al.* JNJ-64041757 (JNJ-757), a Live, Attenuated, Double-Deleted *Listeria monocytogenes*-Based Immunotherapy in Patients With NSCLC: Results From Two Phase 1 Studies. *JTO Clin Res Rep* **2**, 100103 (2021).
8. Toso, J. F. *et al.* Phase I Study of the Intravenous Administration of Attenuated *Salmonella typhimurium* to Patients With Metastatic Melanoma. *Journal of Clinical Oncology* **20**, 142–152 (2002).
9. Neuwelt, A. J. *et al.* Cancer cell-intrinsic expression of MHC II in lung cancer cell lines is actively restricted by MEK/ERK signaling and epigenetic mechanisms. *J Immunother Cancer* **8**, e000441 (2020).
10. Biot, C. *et al.* Preexisting BCG-specific T cells improve intravesical immunotherapy for bladder cancer. *Sci Transl Med* **4**, (2012).
11. Nemes, E. *et al.* Prevention of *M. tuberculosis* Infection with H4:IC31 Vaccine or BCG Revaccination. *New England Journal of Medicine* **379**, 138–149 (2018).
12. Böttcher, J. P. *et al.* NK Cells Stimulate Recruitment of cDC1 into the Tumor Microenvironment Promoting Cancer Immune Control. *Cell* **172**, 1022–1037.e14 (2018).
13. Liu, P. *et al.* Crizotinib-induced immunogenic cell death in non-small cell lung cancer. *Nat Commun* **10**, 1486 (2019).

REVIEWERS' COMMENTS

Reviewer #1 (expert in BCG in autoimmune disease):

Absent.

Reviewer #2 (expert in tumour Immunology and immune checkpoint therapy):

The authors have properly addressed most of my questions in the revised version of their manuscript. Several figures have been included in the main manuscript and the supplementary data. The authors also provided a meaningful discussion in the rebuttal.

Reviewer #3 (expert in thoracic Oncology and NSCLC):

The authors have done an excellent job of not only responding effectively to the concerns raised during my initial review, but also those of Reviewer 1 and 2. This revised submission now more accurately presents and contextualizes the central conclusions of the research by adding additional lung cancer models and also effectively justifying the use of the melanoma models used. Further, significant effort has been made to improve the clarity and readability of this revised submission. For these reasons, I do support publication of this revised manuscript in Nature Communications.

POINT-BY-POINT REPLY LETTER.
MANUSCRIPT NUMBER NCOMMS-22-39644A.

We appreciate the helpful comments and constructive criticisms of the reviewers.

Next, we will respond **point-by-point** to the different questions and concerns raised by the reviewers (comments in bold and *italics*).

REVIEWER 1

Absent.

REVIEWER 2

The authors have properly adressed most of my questions in the revised version of their manuscript. Several figures have been included in the main manuscript and the supplementary data. The authors also provided a meaningful discussion in the rebuttal.

RESPONSE: We really appreciate the effort of the reviewer to evaluate our study, and we thank him/her for the positive response.

REVIEWER 3

The authors have done an excellent job of not only responding effectively to the concerns raised during my initial review, but also those of Reviewer 1 and 2. This revised submission now more accurately presents and contextualizes the central conclusions of the research by adding additional lung cancer models and also effectively justifying the use of the melanoma models used. Further, significant effort has been made to improve the clarity and readability of this revised submission. For these reasons, I do support publication of this revised manuscript in Nature Communications.

RESPONSE: We really appreciate the effort of the reviewer to evaluate our study, and we thank him/her for the positive feedback to our rebuttal not only to his/her own previous concerns and suggestions, but also to those ones from reviewer 1 and 2.